# Accelerated increases in global and Asian summer monsoon precipitation from future aerosol reductions

Laura J. Wilcox[1,2], Zhen Liu[3], Bjørn H. Samset[4], Ed Hawkins[1,2], Marianne T. Lund[4], Kalle Nordling[5], Sabine Undorf[6], Massimo Bollasina[3], Annica M. L. Ekman[6], Srinath Krishnan[6], Joonas Merikanto[5], and Andrew G. Turner[1,2]

[1]National Centre for Atmospheric Science, UK
[2]Department of Meteorology, University of Reading, Reading, UK
[3]School of Geosciences, Grant Institute, University of Edinburgh, Edinburgh, UK
[4]CICERO Center for International Climate Research, Oslo, Norway
[5]Finnish Meteorological Institute, Helsinki, Finland
[6]Department of Meteorology, Stockholm University, Stockholm, Sweden

**Correspondence:** Laura Wilcox l.j.wilcox@reading.ac.uk

**Abstract.** There is a large range of future aerosol emissions scenarios explored in the Shared Socioeconomic Pathways (SSPs), with plausible pathways spanning a range of possibilities from large global reductions in emissions by 2050 to moderate global increases over the same period. Diversity in emissions across the pathways is particularly large over Asia. Rapid reductions in anthropogenic aerosol and precursor emissions between the present day and the 2050s lead to enhanced increases in global and Asian summer monsoon precipitation relative to scenarios with weak air quality policies. However, the effects of aerosol reductions don't persist to the end of the 21st century for precipitation, when instead the response to greenhouse gases dominates differences across the SSPs. The relative magnitude and spatial distribution of aerosol changes is particularly important for South Asian summer monsoon precipitation changes. Precipitation increases here are initially suppressed in SSPs 2-4.5, 3-7.0, and 5-8.5 relative to SSP1-1.9 when the impact of remote emission decreases is counteracted by continued increases in South Asian emissions.

## 1 Introduction

Anthropogenic aerosols can affect climate either by scattering or absorbing solar radiation, or by changing cloud properties (Boucher et al., 2013). Overall, aerosols have a global-mean cooling effect, manifested, for example, in a slower rate of global warming in the mid twentieth century concurrent with rapid increases in aerosol burden (Wilcox et al. (2013); Jones et al. (2013); Hegerl et al. (2019)). This has raised the question of whether the warming associated with present-day and future reductions in anthropogenic aerosol might exacerbate the climate impacts brought about by continued increases in greenhouse gas (GHG) emissions.

Many studies have demonstrated the potential for an enhanced future warming from aerosol reductions in global climate models driven by plausible reductions in the emissions of anthropogenic aerosol and their precursors (e.g. Chalmers et al. (2012); Levy et al. (2013); Rotstayn et al. (2013); Acosta Navarro et al. (2017)). In recent years, emission scenarios have typically been taken from either the Representative Concentration Pathways (RCPs; Moss et al. (2010); van Vuuren et al. (2011)) used in the 5th Coupled Model Intercomparison Project (CMIP5; Taylor et al. (2012)), or the more diverse ECLIPSE (Klimont et al., 2017) aerosol pathways: CLE (Current LEgislation), and MFR (Maximum Feasible Reductions). Estimates based on transient simulations with CMIP5 generation models suggest that future aerosol reductions may result in warming of up to 1.1K in addition to any GHG-driven warming (e.g. Rotstayn et al. (2013); Levy et al. (2013); Acosta Navarro et al. (2017)). This indicates that reduced anthropogenic aerosol emissions may account for up to half of the total warming by 2100 in scenarios with moderate GHG increases. Similar magnitudes are also seen in studies using equilibrium experiments (Kloster et al., 2010), reduced complexity models (Hienola et al., 2018), and studies assuming a complete removal of anthropogenic aerosol (Samset et al. (2018); Nordling et al. (2019)).

The important role of anthropogenic aerosol in driving precipitation changes has also been documented, including the possible contribution to the spin down in the global water cycle in the mid-twentieth century (Liepert et al. (2004); Wilcox et al. (2013); Wu et al. (2013)). A greater response of global mean precipitation to anthropogenic aerosol changes compared to GHGs is expected since aerosol has a stronger effect on atmospheric shortwave transmissivity, and thus a stronger influence on radiative energy imbalance (e.g. Liepert et al. (2009); Andrews et al. (2010); Rotstayn et al. (2013); Samset et al. (2016); Liu et al. (2018)). The apparent hydrological sensitivity (% change in precipitation divided by absolute change in temperature; Fläschner et al. (2016)) for anthropogenic aerosol is twice that for GHGs (Kloster et al. (2010); Salzmann (2016); Samset et al. (2016)). This enhanced sensitivity means that anthropogenic aerosol reductions might be expected to play a relatively more important role in future increases in global precipitation for a given temperature change. Several studies using CMIP5 models estimate an increase in global mean precipitation between 0.09 and 0.16 mm day$^{-1}$ by 2100 from aerosol reductions (e.g. Levy et al. (2013); Rotstayn et al. (2013); Westervelt et al. (2015)).

The effects of future aerosol reductions are likely to be felt more strongly at regional rather than global scales due to their heterogeneous forcing distribution and strong influence on circulation patterns. Previous work has identified relatively large temperature increases over Europe (Sillmann et al., 2013), the Arctic (Acosta Navarro et al., 2016), and East Asia (Westervelt et al., 2015), compared to the global mean response. For precipitation, the regional response is particularly pronounced for the Asian summer monsoon (Levy et al. (2013); Westervelt et al. (2015); Acosta Navarro et al. (2017); Bartlett et al. (2018); Samset et al. (2018)). Here, precipitation is sensitive to changes in remote aerosol through its control on the interhemispheric temperature gradient and Inter Tropical Convergence Zone (ITCZ) location and the atmospheric wave pattern over Eurasia, and to local aerosol changes, which further modify the local monsoon circulation (Polson et al. (2014); Dong et al. (2016); Guo et al. (2016); Shawki et al. (2018); Undorf et al. (2018)). A number of studies have suggested that historical aerosol increases are a key driver of the observed decrease in Asian summer monsoon precipitation (Lau and Kim (2010); Bollasina et al. (2011); Song et al. (2014); Li et al. (2015); Liu et al. (2019)), and projected increases in precipitation due to aerosol reductions are consistent with this. Importantly, Asia will undergo the largest anticipated future changes in aerosol amounts

worldwide (Lund et al. (2019); Scannell et al. (2019)), and is thus a region likely to see an anthropogenic aerosol influence on near-future precipitation trends.

Despite evidence that anthropogenic aerosols influence temperature and precipitation, quantification of the associated changes is hindered by several compounding uncertainties. The degree by which anthropogenic aerosol reductions enhance future cli-
mate change is model dependent. Models with weaker historical aerosol forcing generally have weaker positive radiative forcing from future aerosol reductions, and therefore predict relatively moderate global mean warming (Gillett and Von Salzen (2013); Westervelt et al. (2015)) and moderate precipitation increases due to future aerosol reductions (Rotstayn et al., 2015), compared to models with larger aerosol forcing. The uncertainty in aerosol radiative forcing itself is currently the largest source of uncertainty in estimates of the magnitude of the total anthropogenic forcing on climate, with the most recent estimate pro-
ducing a 68% confidence interval from -1.60 to -0.65 W m$^{-2}$ (Bellouin et al., 2019). This is comparable to the range simulated by CMIP5 models: -1.55 to -0.68 W m$^{-2}$ (Zelinka et al., 2014). The effect of this uncertainty on regional climate projections may be further enhanced by feedbacks from atmospheric circulation changes (Nordling et al., 2019). The compensating effects from the response to different near-term climate forcers also increases the uncertainty in multi-decadal projections, with future changes in methane and nitrate aerosol having the potential to moderate future temperature enhancements from decreases in
anthropogenic aerosol (Bellouin et al. (2011); Shindell et al. (2012); Pietikäinen et al. (2015)). At regional scales, changes in land use may also play an important role (Singh et al., 2019b).

In opposition to the CMIP5-generation findings summarised above, Shindell and Smith (2019) recently dismissed the possibility that future aerosol reductions might lead to rapid increases in the magnitude or rate of global-mean warming, even in scenarios with aggressive clean air policies, based on simulations with a reduced-complexity impulse response model (Smith
et al., 2018). Yet, this conclusion may not hold when using a fully coupled GCM and when investigating changes beyond global mean temperature. Even if the short atmospheric residence time of anthropogenic aerosol potentially makes their effects negligible on centennial timescales, they are likely important for regional and global climate over the next few decades. This is especially the case for Asia where large aerosol emission changes are anticipated, and where aerosol has played an important role in historical changes, in particular for precipitation. In this study, we examine state-of-the-art models and scenarios in
CMIP6 and make the case for a potential enhancement of increases in global and Asian temperature and precipitation on a 20-30 year time horizon due to removal of anthropogenic aerosol. Such an effect is an important consideration for adaptation and mitigation strategies.

## 2  Data and methods

### 2.1  Models and experiments

We use data from the CMIP6 (Eyring et al., 2016) historical experiment (1850-2014) and four future scenarios following Shared Socioeconomic Pathways (SSPs) 1-1.9, 2-4.5, 3-7.0, and 5-8.5 (O'Neill et al. (2016); Rao et al. (2017); Riahi et al. (2017)), which sample a range of aerosol pathways. At the time of writing, data were only available for between 6 and 14 models for the variables and experiments we consider. The data used in this study are summarised in Table 1. All available

data is used for each experiment, and model means are used in multi-model comparisons, except where otherwise stated. Many CMIP6 models include improved representation of aerosol microphysics and aerosol-cloud interactions compared to CMIP5, such as internal mixing and heterogeneous ice nucleation (e.g. Bellouin et al. (2013); Mulcahy et al. (2018); Kirkevåg et al. (2018); Wyser et al. (2019)), and all models we consider include at least the first aerosol indirect effect (Twomey et al., 1984).

The SSPs used in CMIP6 sample a far greater range of uncertainty in future aerosol and precursor emissions than the RCPs used in CMIP5 (Lund et al. (2019); Scannell et al. (2019)). Partanen et al. (2018) highlighted the importance of uncertainty in aerosol emission pathways for the potential enhancement of global temperature increases from anthropogenic aerosol reductions. The CMIP5 RCPs 2.6-8.5 sampled only a limited range of this emission uncertainty, with an associated difference in global mean temperature of no more than 0.18K throughout the twenty-first century. Contrasting aerosol pathways spanning a
wider range of emission uncertainty resulted in a difference of up to 0.86K (in 2061). The SSPs span most of this wider range of emissions pathways explored by Partanen et al. (2018). They include large, rapid reductions in aerosol and precursor emissions in SSP1-1.9, more moderate reductions (comparable to the RCPs) in SSP2-4.5 and SSP5-8.5, and continued increases in the coming decades in SSP3-7.0 (see Figure 1). Much of the spread in global emission pathways comes from diversity over Asia and North Africa (Lund et al., 2019).

In our analysis, we compare future decadal-mean climate changes to the present day (1980-2014) across SSPs 1-1.9, 2-4.5, 3-7.0, and 5-8.5. We focus on the period up to the 2050s, when aerosol emission uncertainty is largest, but the full range of uncertainty in greenhouse gas (GHG) emissions has yet to emerge (Figure 1). However, the changes in GHG emissions in this period are not negligible, and the emerging modelled climate responses we show include the effects of changes in both anthropogenic aerosols and greenhouse gases. The SSPs also consider a range of land use scenarios, with extensive
and moderate deforestation in SSP3-7.0 and SSP5-8.5 respectively, little change in forest cover in SSP2-4.5 before 2050, and large-scale afforestation in SSP1-1.9 (O'Neill et al., 2016). Where data are available, we have provided the global-mean annual-mean effective radiative forcings (ERFs) due to historical changes in anthropogenic aerosols, GHGs, and land use changes in Table 2, as an indicator of the relative importance of these changes on centennial timescales. Typically GHG forcing over the historical period is 2-3 times larger than aerosol forcing, which is in turn an order of magnitude greater than land use forcing.
When considering the Asian summer monsoon, the regional pattern of forcing is also important (e.g. Dong et al. (2019)). The CMIP6 mean ERF over the Asian region is shown in Figure 2 for anthropogenic aerosols, GHGs, land use and land cover change, and total anthropogenic drivers (ERF from individual models is shown in Supplementary Figures 1-4 for each driver). Anthropogenic aerosols and GHGS are the main contributors to historical anthropogenic forcing over Asia (Figure 2).

Historical ERF can only be a first-order indicator of the potential relative importance of each driver in future changes, as the
distribution and magnitude of future changes may differ, especially for non-GHG forcing. However, comparison of the extreme SSPs for anthropogenic aerosol and land use and land cover changes in 2050 shows that the magnitude of changes between the present day and 2050 is roughly comparable to to the changes over Asia between 1850 and 2014 (Supplementary Figure 5). This suggests that the historical ERFs are a good indicator of the relative magnitude of these forcings in future, and that GHG and anthropogenic aerosol changes are likely to remain the main drivers of Asian summer monsoon trends in the future. Thus,
we focus on the effects of anthropogenic aerosol and GHG in our analysis.

Since multiple forcing agents vary simultaneously in the SSPs, it is not possible to quantify the respective effects of aerosol and GHG changes, although the main driver of an anomaly can still be identified. Consider the global emission changes shown in Figure 1. SSP1-1.9 has the largest aerosol and GHG emission reduction, while SSP3-7.0 has a moderate increase in aerosol emissions (reverse climate response to SSP1-1.9) and moderate increase in GHG emission (enhanced climate response to SSP1-1.9). If the magnitude of the climate response to these changes decreases monotonically from SSP1-1.9 to SSP3-7.0 (Figure 3), this indicates that aerosol changes are the main driver of the climate response. Further confirmation of aerosol as the main driver is gained from the comparison of the anomalies in SSP2-4.5 and SSP5-8.5, which have similar aerosol pathways, but very different greenhouse gas pathways (Figure 1, Figure 3). If the climate response in these two scenarios is similar, then the greenhouse gas influence has yet to emerge over the aerosol signal. As the differences in greenhouse gas emissions between the two scenarios increase, a larger response is expected in SSP5-8.5, which has large increases in global GHG emissions compared to very moderate increases in SSP2-4.5 (Figure 1). In cases where GHGs are the main driver of the response, the magnitude will increase monotonically from SSP1-1.9 to SSP5-8.5 (Figure 3).

In Section 4.1, we use an additional DAMIP (Detection and Attribution Model Intercomparison Project; Gillett et al. (2016)) experiment, SSP2-4.5-aer. This differs from the companion SSP2-4.5 in that only aerosol emissions are evolving while all other forcings are held constant at their 1850 levels. This scenario allows the response to anthropogenic aerosols to be seen in isolation from the response to greenhouse gas changes and may thus provide support to any conclusion drawn from the analysis of the SSP2-4.5 experiment. Data for this experiment is so far only available for two models, CanESM5 and MIROC6 (Shiogama (2019); Swart et al. (2019a)). In this analysis, decadal mean anomalies are again presented relative to the present day (1980-2014). However, in this case, the present day is necessarily defined based on the historical-aer simulation (a historical simulation where only anthropogenic aerosol and precursor emissions are transient, also included in DAMIP).

## 2.2 Present day model evaluation

Here, we use a number of observation and reanalysis datasets to present a broad evaluation of the performance of CMIP6 models in reproducing present day (1980-2014) climatologies and linear trends in global temperature and precipitation, the interhemispheric temperature gradient, and the Asian summer monsoon. Global temperature observations are taken from GIS-TEMP v4 (Hansen et al. (2010); Lenssen et al. (2019)), the Goddard Institute for Space Studies gridded dataset, which is based on GHCN v4 over land (Global Historical Climatology Network; Menne et al. (2018)) and ERSST v5 over ocean (Extended Reconstructed Sea Surface Temperature; Huang et al. (2017)). GISTEMP is provided as anomalies relative to 1951-1980 on a $2° \times 2°$ grid. For global precipitation, data from the Global Precipitation Climatology Project (GPCP; Adler et al. (2003)) are used on a $2.5° \times 2.5°$ grid. GPCP combines gauge- and satellite-based observations over land with satellite observations over ocean. Since there can be large discrepancies between precipitation observations from different sources (Collins et al. (2013); Sperber et al. (2013); Prakash et al. (2015)), we use a number of datasets in our evaluation of the Asian summer monsoon. Precipitation observations over land are also taken from APHRODITE (Asian Precipitation - Highly-Resolved Observational Data Integration Towards Evaluation; Yatagai et al. (2012)) and the Global Precipitation Climatology Centre (GPCC; Schneider et al. (2014)). APHRODITE contains data from a dense network of rain gauges and is used at $0.25° \times 0.25°$ resolution,

within the domain bounded by 60°E, 150°E, 15°S, and 55°N. GPCC also provides gauge-based data, but at a reduced horizontal resolution ($0.5° \times 0.5°$) compared to APHRODITE. We also show precipitation from CMAP (Climate Prediction Centre Merged Analysis of Precipitation; Xie et al. (1996); Xie et al. (1997)), which blends satellite and gauge-based estimates with NCEP/NCAR reanalysis precipitation. The atmospheric circulation plays an important role in the distribution of Asian summer monsoon precipitation, so we also compare upper- and lower-tropospheric winds from CMIP6 models to ERA-Interim (Dee et al., 2011).

The global-mean annual-mean temperature anomaly from GISTEMP falls within the range of the CMIP6 ensemble during the historical period (Figure 4a). However, most models overestimate the rate of recent warming (Figure 4a, e). The inter-hemispheric temperature gradient (Northern Hemisphere - Southern Hemisphere) anomalies are also consistent in GISTEMP and CMIP6 (Figure 4b), although the models generally have anomalies that are more positive than seen in the observations. Most models reproduce the negative trend in the interhemispheric temperature gradient in 1950-1974 (Figure 4e), which is associated with a global increase in anthropogenic aerosol and a weakening of the global monsoon (e.g. Polson et al. (2014)). Most models also capture the positive trend in interhemispheric temperature gradient since 1980-2014 when rates of change in the global aerosol burden were relatively small (Figure 4e).

All models simulate an increase in global-mean annual mean precipitation since 1980, and most model members simulate larger trends than observations (Figure 4c,e). The observed trend in Asian (67.5-145°E, 5-47.5°N) summer (June-August, JJA) precipitation is small compared to interannual variability, and this is reflected in large uncertainty in the sign of the modelled trend (Figure 4d, e).

Compared to APHRODITE, the CMIP6 ensemble mean underestimates summer monsoon precipitation amount over India, and overestimates it over the Tibetan Plateau and the Indochina peninsula (Figure 5a, c, e). The magnitude of the bias between the CMIP6 multi-model mean and APHRODITE is comparable to the magnitude of the difference between observational datasets over northeast China and India, but the model biases over the Tibetan Plateau and the Indochina peninsula are relatively large (Figure 5d, e). The modelled meridional component of the monsoon circulation at 850hPa is too strong over the Equatorial Indian Ocean, while the flow over the Bay of Bengal, and the extension of the circulation into China, is too weak (Figure 5e). This pattern is seen in almost all models, and is highlighted in the comparison of the zonal and meridional components of the 850 hPa wind in CMIP6 and ERA-Interim in Figure 5f. This weak extension of the summer monsoon into eastern China, with an anomalously strong extension into the subtropical west Pacific (Figure 5e), is consistent with the pattern of differences between CMIP5 models and observations (Sperber et al., 2013). Nevertheless, the CMIP6 models are more skilful in their representation of the Indian summer monsoon compared to CMIP5 (Gusain et al., 2020).

While aspects of the CMIP6 multi-model mean summer monsoon compare well with observations, and the multi-model mean performs better than the individual models, there is a large inter-model diversity in the monsoon precipitation and atmospheric circulation (summarised in Figure 5f; maps of present-day means, and anomalies compared to both APHRODITE and GPCP are shown for individual models in Supplementary Figures 6-10). Of the models we will consider on an individual basis in Section 4.1, CanESM5 has a small regional mean precipitation bias, but a weak pattern correlation, compared to APHRODITE. It has a particularly large dry bias over India, with less than 3 mm day$^{-1}$ in the seasonal mean, and a relatively

large excess of precipitation over the Tibetan Plateau and into China. MIROC6 performs relatively well over land, but has excessive precipitation west of India (Supplementary Figures 6, 7, 9, 10). Such biases may affect the pattern of the precipitation anomaly in the SSPs relative to the present day (Wilcox et al., 2015).

Aerosol optical depth (AOD) is a measure of the extinction of solar radiation due to scattering and absorption by an aerosol layer. Comparison of 550nm AOD from CMIP6 and MODIS (Remer et al. (2008); Platnick (2015)) for the common 2002-2014 period shows that models underestimate AOD over much of the NH outside Asia, and overestimate it in the SH midlatitudes (Figure 6a-c). This pattern is common across models, with the some exceptions over Asia (Supplementary Figure 11 shows the comparison between AOD from individual models and MODIS). CanESM5 overestimates AOD over Eurasia, and northern China in particular, compared to MODIS. UKESM1-0-LL and IPSL-CM6A-LR have more moderate overestimates of AOD over parts of Asia. However, both models underestimate AOD over eastern China, where very high AOD is seen in MODIS.

## 2.3 Future anthropogenic aerosol changes

In all SSPs the largest AOD changes are over Asia (Figure 6d-o), consistent with the changes in aerosol and precursor emissions (Figure 1). Future changes in AOD are characterised by global decreases in SSP1-1.9, with the exception of an initial increase over South Asia, and positive anomalies relative to 1980-2014 over the Tibetan Plateau and southern Africa, which may have a dust component (Figure 6d-f). AOD changes are characterised by regional contrasts in SSP2-4.5 and SSP5-8.5, with an overall decrease in the NH contrasted with an increase in the SH, and large decreases over East Asia against large increases over South Asia until the 2030s (Figure 6g-i, m-o). This Asian dipole pattern is persistent in SSP2-4.5 and SSP5-8.5 and strengthens from the present until the 2040s. In these pathways, aerosol and precursor emission are similar, with the large increases in South Asian AOD predominantly driven by increases in $SO_2$ emissions (Figure 1). Emissions of black carbon in South Asia do follow different trajectories in the two pathways, but black carbon accounts for a smaller proportion of the total emission (Figure 1), and thus the total AOD. The final scenario we consider, SSP3-7.0, again contrasts widespread decreases in NH AOD against increases in the SH (Figure 6j-l). However, this scenario also includes large aerosol and precursor emission and AOD increases over East Asia and particularly South Asia. These increases are driven predominantly by $SO_2$ over South Asia, but have a BC contribution over East Asia (Figure 1). As East Asian $SO_2$ is roughly constant between 2014 and 2050 in SSP3-7.0, much of the large positive anomaly there is a reflection of the large positive trend in AOD between 1980 and 2014. The AOD pattern in SSP3-7.0 persists through the three periods shown in Figure 6, but the East Asian increase starts to weaken by 2050 (Figure 1).

## 3 Global response

Global-mean annual-mean temperature, precipitation, interhemispheric temperature gradient, and hydrological sensitivity anomalies for 2025-2034, 2035-2044, and 2045-2054 relative to 1980-2014 are shown in Figure 7. The boxes show the interquartile range, based on the mean response from each individual model. Individual model responses are overlaid as black diamonds, and the horizontal bar shows the multi-model median. The 95% confidence interval (95% CI) about a median can be found

from the empirical relationship:

$$95\%CI = \pm\frac{1.57 \times IQR}{\sqrt{n}} \tag{1}$$

where IQR is the interquartile range and n is the number of points (McGill et al., 1978). For the sample size used in this work (n=11), $\frac{1.57}{\sqrt{n}} \approx 0.5$, so we use the interquartile range to determine significance, rather than assuming that the confidence interval is symmetric about the median. Significant differences between anomalies from different SSPs are identified when the median anomaly from one SSP lies outside the interquartile range of another, and give additional support to the qualitative aerosol- and GHG-driven patterns sketched in Figure 3.

For each period shown in Figure 7a, the global mean temperature anomalies are broadly ordered according to their GHG pathway, and diverge with time in a similar fashion to GHG emissions (Figure 1c, Figure 3). By 2045-2054, SSP2-4.5, 3-7.0, and 5-8.5 anomalies relative to 1980-2014 are significantly larger than those for SSP1-1.9. The anomaly from SSP5-8.5 is also significantly larger than that from SSP2-4.5. This suggests that anthropogenic aerosol plays a limited role in the evolution of global mean near-surface temperature on these timescales, supporting the conclusions of Shindell and Smith (2019). However, as will be discussed in Section 4, anthropogenic aerosol does play a role in the pattern and magnitude of regional temperature change. Importantly, anthropogenic aerosol is the main driver of trends in the interhemispheric temperature gradient until 2050 (Figure 7d, Figure 3), which has a strong control on ITCZ position and the global monsoon, and thus regional precipitation. There is a large spread in interhemispheric temperature gradient anomalies across models, consistent with the large uncertainty in historical trends (Figure 4e), but a monotonic increase in the magnitude of the median anomaly from SSP3-7.0 to SSP1-1.9, consistent with an aerosol-driven response, is present in all three future periods. In 2035-2044 and 2045-2054 the SSP3-7.0 anomaly is significantly smaller than the SSP1-1.9 anomaly. The dominant role of anthropogenic aerosol is further supported by the comparable magnitude of anomalies in SSP2-4.5 and SSP5-8.5 (Figure 7d, Figure 3).

There is a clear aerosol-driven signal in future increases in global mean precipitation and hydrological sensitivity (Figure 7b, c, Figure 3), with a significantly larger median anomaly in SSP1-1.9 compared to SSP3-7.0 in 2025-2034 and 2035-2044. There is the suggestion of the beginning of a shift towards GHGs as the dominant driver of precipitation increases in 2050, where the median SSP1-1.9 anomaly is marginally smaller than in SSP2-4.5, and the median anomaly in SSP5-8.5 is marginally larger than that in SSP2-4.5. GHGs are the main driver of global precipitation change by the end of the 21st century (not shown). The aerosol signal in hydrological sensitivity is larger and more persistent. SSP1-1.9 has significantly larger median hydrological sensitivity anomalies relative to 1980-2014 than SSP3-7.0 in 2025-2034, and significantly larger median anomalies than all other SSPs in 2035-2044 and 2045-2054. SSP1-1.9 anomalies remain larger throughout the 21st century (not shown).

A number of prominent outliers can be seen in Figure 7. These points are not indicators of uncertainty in the response to anthropogenic aerosol emissions in the SSPs, or in the relative differences in the anomalies compared to 1980-2014 from each SSP: for each variable the outlying model is the same for each period; and for precipitation and hydrological sensitivity the high outliers can be seen to show the same aerosol-driven pattern across the periods as the multi-model median response. The outliers are likely reflections of differing climate sensitivities in the models (Table 2). As shown in Figure 4e, there are large positive trends in both global mean temperature and precipitation between 1980 and 2014, which contribute to the magnitude

of the anomalies shown in Figure 7. Models in Figure 7 with large temperature anomalies are CanESM5 and UKESM1-0-LL. MIROC6 has the smallest anomalies. These models are also those with the highest and lowest equilibrium climate sensitivities in our ensemble (Table 2). For precipitation, the large outlier is UKESM1-0-LL and the small outlier is CAMS-CSM1-0. The precipitation climatologies and patterns of future changes for these models are not unusual compared to other models, as can
be seen in Supplementary Figures 6-10, and 16.

## 4   Asian summer monsoon response

The decrease in Asian monsoon precipitation observed in the second half of the twentieth century has been largely attributed to the global increase in anthropogenic aerosols (Bollasina et al. (2011); Song et al. (2014); Polson et al. (2014)). The hemispheric asymmetry in aerosol forcing leads to an energy imbalance between the hemispheres, which in turn causes a slowdown of the
meridional overturning circulation, and a weakening of the monsoon circulation (Bollasina et al. (2011); Song et al. (2014); Lau et al. (2017); Undorf et al. (2018)). Local aerosol emissions further modify monsoon circulation and precipitation (Cowan and Cai (2011); Guo et al. (2015); Undorf et al. (2018)). In contrast to anthropogenic aerosols, where circulation changes are an important component of the response to forcing, GHGs mainly affect (increase) monsoon precipitation by enhancing tropospheric water vapour, and thus increasing moisture transport toward India (Li et al., 2015).
Global aerosol reductions in SSP1-1.9 briefly cause faster warming over all Asian regions than the other scenarios considered, with SSP1-1.9 warming significantly more than SSP3-7.0, but this effect does not persist beyond the 2040s (Figure 8a). However, anthropogenic aerosol does affect the regional pattern of warming (Figure 9), with slower increases in land temperature in many areas, and India in particular, in SSP2-4.5, 3-7.0, and 5-8.5 compared to SSP1-1.9. This pattern is robust across models (Supplementary Figure 12). The growing influence of GHGs with time can also be seen in Figure 8a and 9 as greater
warming in SSP5-8.5 compared to SSP2-4.5 in the 2040s. The GHG-driven pattern is established by 2045-2054, when all SSPs warm significantly more than SSP1-1.9. Continued increases in anthropogenic aerosol emissions in SSP3-7.0 appear to moderate land warming compared to other SSPs, despite large GHG increases (Figures 8a and 9).

The influence of aerosol is more clearly seen, and more persistent in time, in regional mean precipitation than regional mean temperature (Figure 8a, b; Figure 3), as for the global mean case (Figure 7b). Over Asia, the largest model-median precipitation
increase relative to 1980-2014 occurs in SSP1-1.9 for 2025-2034 and 2035-2044 (significantly larger than SSP2-4.5 in 2025-2034, and significantly larger than SSP3-7.0 in all periods shown). The smallest precipitation increases are seen in SSP3-7.0 during these periods. Increases in SSP2-4.5 lie between those in SSP3-7.0 and SSP1-1.9. The same pattern is seen over East Asia. There is some indication that precipitation anomalies relative to 1980-2014 are slightly larger in SSP5-8.5 compared to SSP2-4.5, but the growing difference between the two scenarios that was seen for temperature is not seen here. By 2100, GHGs
are the dominant influence on the relative magnitude of the future increases in Asian summer monsoon precipitation across the SSPs, but the timing of the transition is model-dependent, as illustrated in Figure 10.

Over Asia and East Asia, precipitation increases relative to 1980-2014 are significantly smaller in SSP3-7.0 compared to SSP1-1.9 until the mid-21st century (Figure 8b). Over East Asia, JJA mean precipitation is not significantly larger than in

1980-2014 in SSP3-7.0 until 2045-2054. A similar pattern of aerosol-dominated differences between the SSPs is seen in hydrological sensitivity over Asia and East Asia (Figure 8c; Figure 3). The beginnings of the GHG influence are seen in hydrological sensitivity over East Asia in 2045-2054, when the SSP5-8.5 anomaly is slightly larger than that from SSP2-4.5. However, the clear GHG-dominated pattern seen for temperature (Figure 8a) is not seen here.

5  Figures 11 and 12 show that the pattern of Asian precipitation changes are similar, regardless of the emission pathway that is followed, but that the magnitude of the changes are pathway dependent, as summarised in Figure 8. Figure 11 shows the absolute anomaly compared to 1980-2014, while Figure 12 shows the anomaly relative to the SSP1-1.9 response. In Figure 12, the influence of GHGs can be seen, with greater GHG emissions driving greater drying over the Equatorial Indian Ocean and further increases in precipitation over India (particularly in the comparison between SSP2-4.5 and SSP5-8.5, and in SSP3-7.0

10 in the 2050s), as for temperature (Figure 9). Precipitation increases are smaller in SSP2-4.5, SSP3-7.0, and SSP5-8.5 compared to SSP1-1.9 (Figure 8, 11), mainly due to differences over northern India, Bangladesh, and the Bay of Bengal (Figure 12). The pattern seen in the model-mean anomaly (Figure 12) is robust across models (Supplementary Figure 13).

  The pattern of precipitation anomalies relative to 1980-2014 across the SSPs is different over South Asia compared to East Asia and Asia (Figure 8, Figure 3). This suggests that the continued increases in local aerosol emissions may be relatively

15 more important here than the remote decreases. All precipitation anomalies are positive, although many are not significantly different to zero. In 2025-2034 and 2035-2044 precipitation and hydrological sensitivity anomalies in SSP2-4.5 and SSP5-8.5 are smaller than those in SSP1-1.9 and SSP3-7.0, following neither the pattern expected from global and East Asian aerosol pathways, nor the GHG pathways (Figure 3). This similarity between SSP2-4.5 and SSP5-8.5, which is seen in all three periods for South Asia, suggests that the dipole in aerosol and precursor emissions and AOD anomalies between East Asia and South

20 Asia in these scenarios (Figure 1; Figure 6) may be further suppressing future increases in precipitation over South Asia due to feedback between the East and South Asian monsoon system responses to forcing (Ha et al. (2018); Singh et al. (2019a)). Overall, there is much more uncertainty in the South Asian precipitation changes compared to East Asia, as evidenced by the larger inter-model spread (Figure 8). Land use changes may also contribute to scenario uncertainty in South Asian precipitation changes (Singh et al., 2019b), although they are likely to be very model-dependent, and influence smaller spatial scales than

25 those we consider here (Figure 2; Supplementary Figure 3).

## 4.1 Aerosol only SSP2-4.5

The deviation of the South Asian precipitation response from the GHG- and aerosol-driven patterns sketched in Figure 3 suggests that the response to local aerosol changes, which have a different time evolution to those over East Asia and in the global mean (Figure 1), may be competing with the response to remote aerosols emission changes. Such behaviour is consistent

30 with the findings in earlier investigations of precipitation changes over South Asia, which have shown that both regional and local changes in anthropogenic aerosol are required to reproduce historical trends (e.g. Guo et al. (2016); Undorf et al. (2018)). The suppressed precipitation increase in SSP2-4.5 and SSP5-8.5 relative to SSP1-1.9 and SSP3-7.0 is a fingerprint of the Asian dipole in AOD trends seen in these scenarios until the mid-twenty-first century, and in current observations (Figure 6; Samset et al. (2019)).

Analysis of SSP2-4.5-aer, in which only anthropogenic aerosol emissions are varying with time following the SSP2-4.5 pathway, allows the response to aerosol changes in this pathway to be isolated from that due to GHG increases. In this case, a dipole in temperature anomalies, with cooling over India and warming over East Asia, and in sea-level pressure, with a positive anomaly over India, the Bay of Bengal, and the Indochina peninsula, and a negative anomaly over the rest of Asia, can clearly be seen in both MIROC6 (Figure 13) and CanESM5 (Supplementary Figure 14). This feature matches the dipole pattern in AOD changes (Figure 6), and is apparent in the SSP2-4.5 response up to 2050 as a moderated GHG-induced warming over South Asian relative to East Asia (Figure 9, Figure 14). Comparing the SSP2-4.5aer and SSP2-4.5 responses (Figure 13 vs. Figure 14 for MIROC6; Supplementary Figure 14 vs. 15 for CanESM5) shows that aerosol largely acts to offset the GHG-driven response, rather than determining the overall pattern of the response.

Differences in the character of the precipitation anomaly can be seen when comparing the anomalies pre- and post-2050. In the earlier period, when South Asian aerosol emissions continue to increase, precipitation anomalies are either weakly positive or negative over India, the Bay of Bengal, and the Indochina peninsula. Post-2050, when anthropogenic aerosols are decreasing throughout Asia, precipitation increases are larger relative to 1980-2014. There are also suggestions of this structure in the CMIP6 mean SSP2-4.5 response, where increases in precipitation are weak over India and the Bay of Bengal compared to the SSP1-1.9 response (Figure 11, 12), but it is not as clear. This is likely partly due to the influence of GHG increases, and partly due to the effects of taking the mean response over models with large differences in their mean precipitation field (Figure 5). However, a number of the individual models do simulate a similar tripolar pattern in precipitation change in SSP2-4.5 to those in MIROC6 (Figure 11; Figure 12; Supplementary Figure 9). Further study into the precipitation responses to the Asian aerosol dipole is needed to understand the mechanisms underlying this response.

## 5 Conclusions

There is large uncertainty in future anthropogenic aerosol emission pathways. This is likely to be of limited importance for global-mean temperature, but anthropogenic aerosol does play an important role in changes in regional temperature and global and regional precipitation until 2050 under the Shared Socioeconomic Pathways. Rapid reductions in anthropogenic aerosol and precursor emissions in SSP1-1.9 lead to larger increases in global and Asian summer monsoon precipitation compared to SSP2-4.5, 3-7.0, and 5-8.5 over East Asia and South Asia, despite the large decrease in greenhouse gases in SSP1-1.9.

In SSP2-4.5, 3.-7.0, and 5-8.5, anthropogenic aerosol emissions continue to increase until the mid-21st century over South Asia. This leads to weaker future increases in regional precipitation until 2050, compared to SSP1-1.9, particularly over northern India. In SSP2-4.5 and 5-8.5 continued increases in South Asian aerosol optical depth occur while it decreases over East Asia. This may further suppress the precipitation increases in northern India compared to SSP1-1.9. However, there is large inter-model uncertainty in the South Asian precipitation changes.

A dipole in aerosol optical depth trends over Asia has been observed since 2010 (Samset et al., 2019), suggesting that SSP2-4.5, SSP5-8.5 (where the current AOD dipole pattern persists), or SSP1-1.9 (where anthropogenic aerosol is reduced in both regions), are more likely to be followed in the real world than SSP3-7.0 (where anthropogenic aerosol increases in both

regions). This presents the possibility of large uncertainty in South Asian summer monsoon precipitation on a 30-50 year time horizon due to uncertainty in local aerosol emission pathways.

*Data availability.* All data used in this work are freely available for research purposes.

*Author contributions.* All authors designed the analysis and wrote the paper. LJW, ZL, BHS, EH, MTL, KN, and SU performed the analysis.

*Competing interests.* The authors declare that they have no conflict of interest.

*Acknowledgements.* This work and its contributors Laura Wilcox, Zhen Liu, and Massimo Bollasina were supported by the UK-China Research & Innovation Partnership Fund through the Met Office Climate Science for Service Partnership (CSSP) China as part of the Newton Fund. Laura Wilcox received additional support from the Natural Environment Research Council (NERC; grant NE/S004890/1, EMERGENCE) and the International Meteorological Institute (IMI) visiting scientist program. Marianne T. Lund and Bjørn H. Samset
acknowledge funding by the Research Council of Norway through grant no. 248834 (QUISARC). Kalle Nordling and Joonas Merikanto acknowledge support from the Academy of Finland project RECIA (grant no. 287440) and European Research Council project ECLAIR (grant no. 646857).

We acknowledge the World Climate Research Programme, which, through its Working Group on Coupled Modelling, coordinated and promoted CMIP6. We thank the climate modelling groups for producing and making available their model output, the Earth System Grid Fed-
eration (ESGF) for archiving the data and providing access, and the multiple funding agencies who support CMIP6 and ESGF. APHRODITE precipitation data are available from http://www.chikyu.ac.jp/precip. GISTEMP temperature data, and CMAP, GPCC, and GPCP precipitation data were provided by the NOAA/OAR/ESRL PSD, Boulder, Colorado, USA, from their website at https://www.esrl.noaa.gov/psd/. We also acknowledge the use of ERA-Interim data produced by ECMWF and provided by the British Atmospheric Data Centre and the National Centre for Atmospheric Science. The analysis in this work was performed on the JASMIN super-data-cluster (Lawrence et al., 2012).
JASMIN is managed and delivered by the UK Science and Technology Facilities Council (STFC) Centre for Environmental Data Archival (CEDA).

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

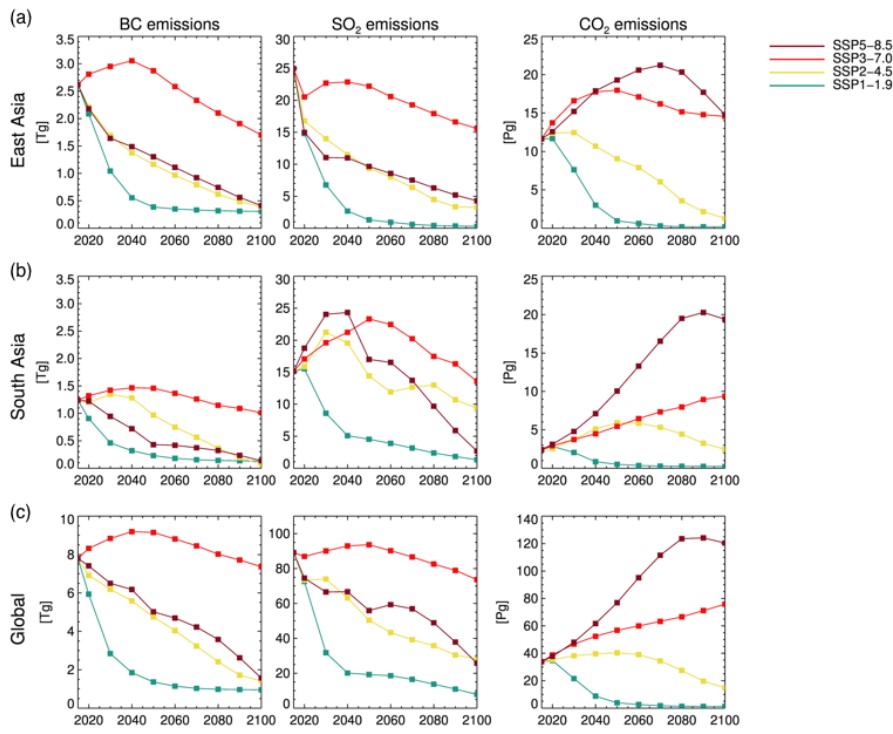

**Figure 1.** (a): Black carbon [Tg], (b): sulphur dioxide [Tg], and (c): carbon dioxide emissions [Pg] over East Asia for SSPs 1-19, 2-45, 3-70, and 5-85. (d)-(f): emissions over South Asia. (g)-(h): global total emissions. East Asia is the region from 20-40°N and 100-120°E. South Asia is the region from 5-25°N and 55-95°E.

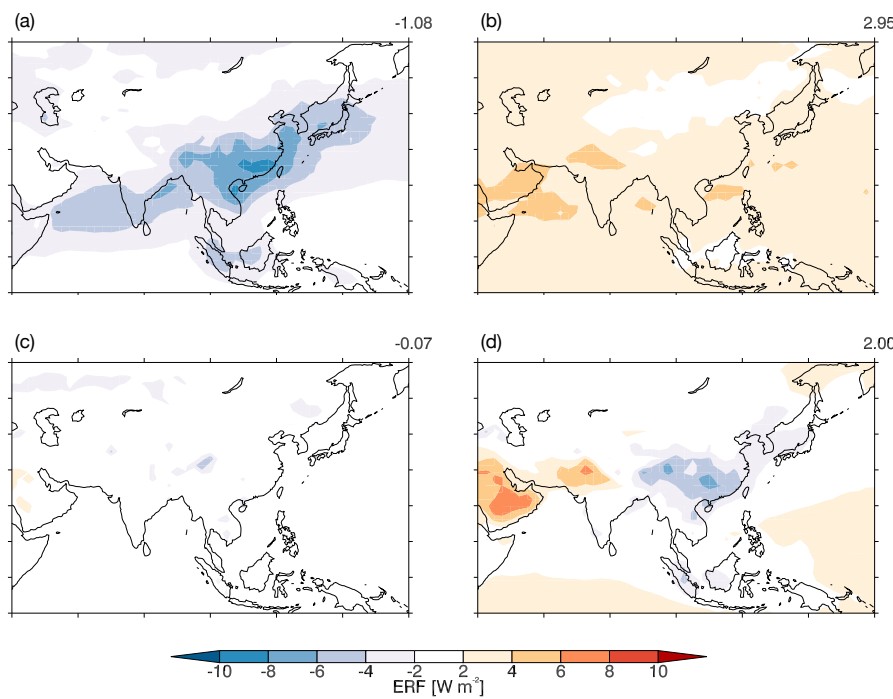

**Figure 2.** CMIP6 historical (2014 vs. 1850) Effective Radiative Forcing due to (a): anthropogenic aerosols, (b): greenhouse gases, (c): land use and land cover changes, and (d): all anthropogenic drivers, calculated based on the models highlighted in Table 2. The number in the top right of each panel shows the global mean value [W m$^{-2}$].

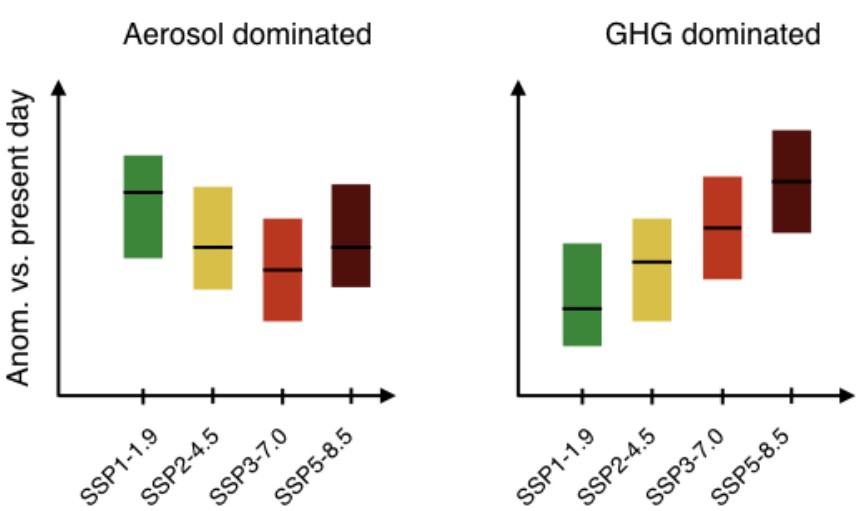

**Figure 3.** Schematics showing the anticipated pattern of anomalies relative to 1980-2014 across the SSPs in cases where the differences are aerosol-dominated and GHG-dominated, based on the global emission pathways shown in Figure 1c. Boxes show the interquartile range of CMIP6 models, black lines show the median. Significant differences between the SSPs are seen when the median from one SSP falls outside the interquartile range of another. Significant differences between SSPs are not a condition for identification of the respective patterns, but may be observed between the extreme scenarios in each case.

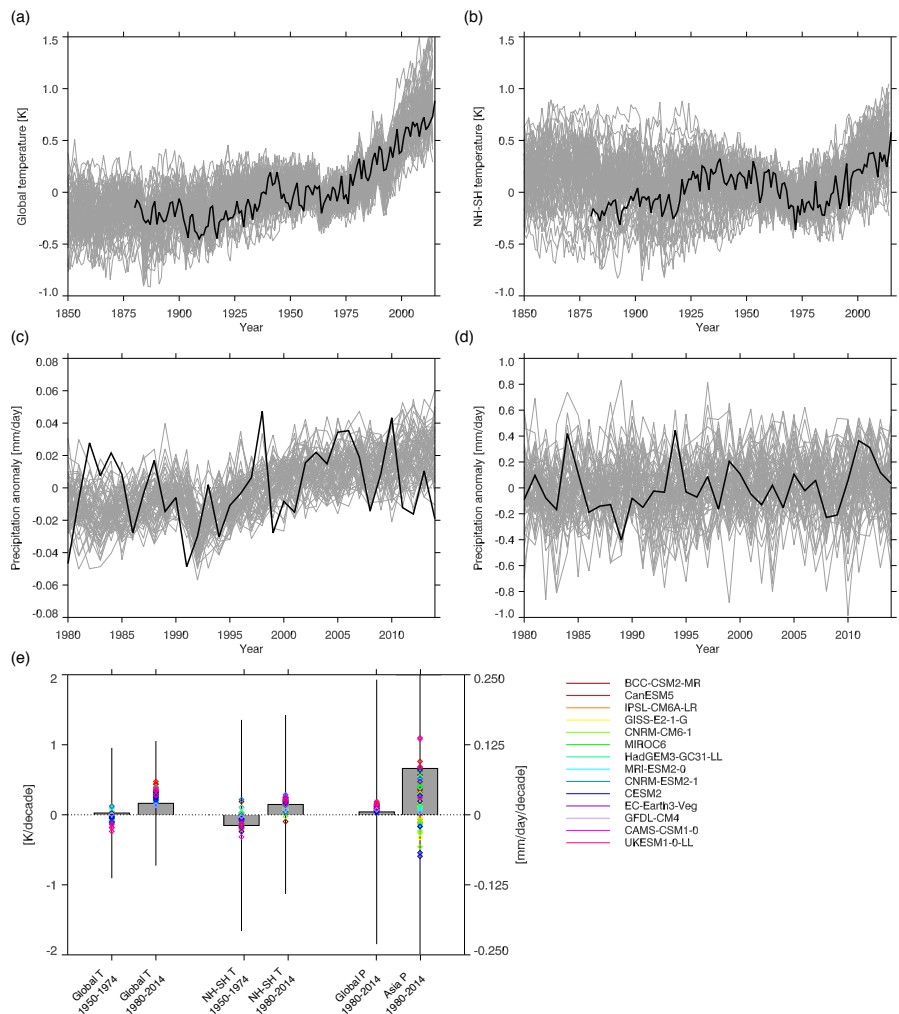

**Figure 4.** (a): Global-mean annual-mean temperature anomaly relative to 1951-1980 from CMIP6 (grey lines show individual members) and GISTEMP (black). (b): Annual-mean interhemispheric temperature gradient anomaly relative to 1951-1980 from CMIP6 (grey) and GISTEMP. (c): Annual-mean global-mean precipitation anomaly relative to 1980-2014 from CMIP6 (grey) and GPCP (black). (d): JJA mean Asia-mean precipitation anomaly relative to 1980-2014 from CMIP6 (grey) and GPCP (black). Asia is the region from 5-47.5°N and 67.5-145°E. (e): linear trends in annual-mean global-mean temperature and JJA-mean interhemispheric temperature gradient from CMIP6 (coloured diamonds) and GISTEMP (grey bars) for 1950-1974 and 1980-2014, and linear trends in annual-mean global-mean precipitation and JJA-mean Asia-mean precipitation for 1980-2014 (grey bars are GPCP). Error bars show plus or minus one standard error on the observed trend. Note that for Asian precipitation this extends beyond the range of the plot, and is an order of magnitude larger than the trend.

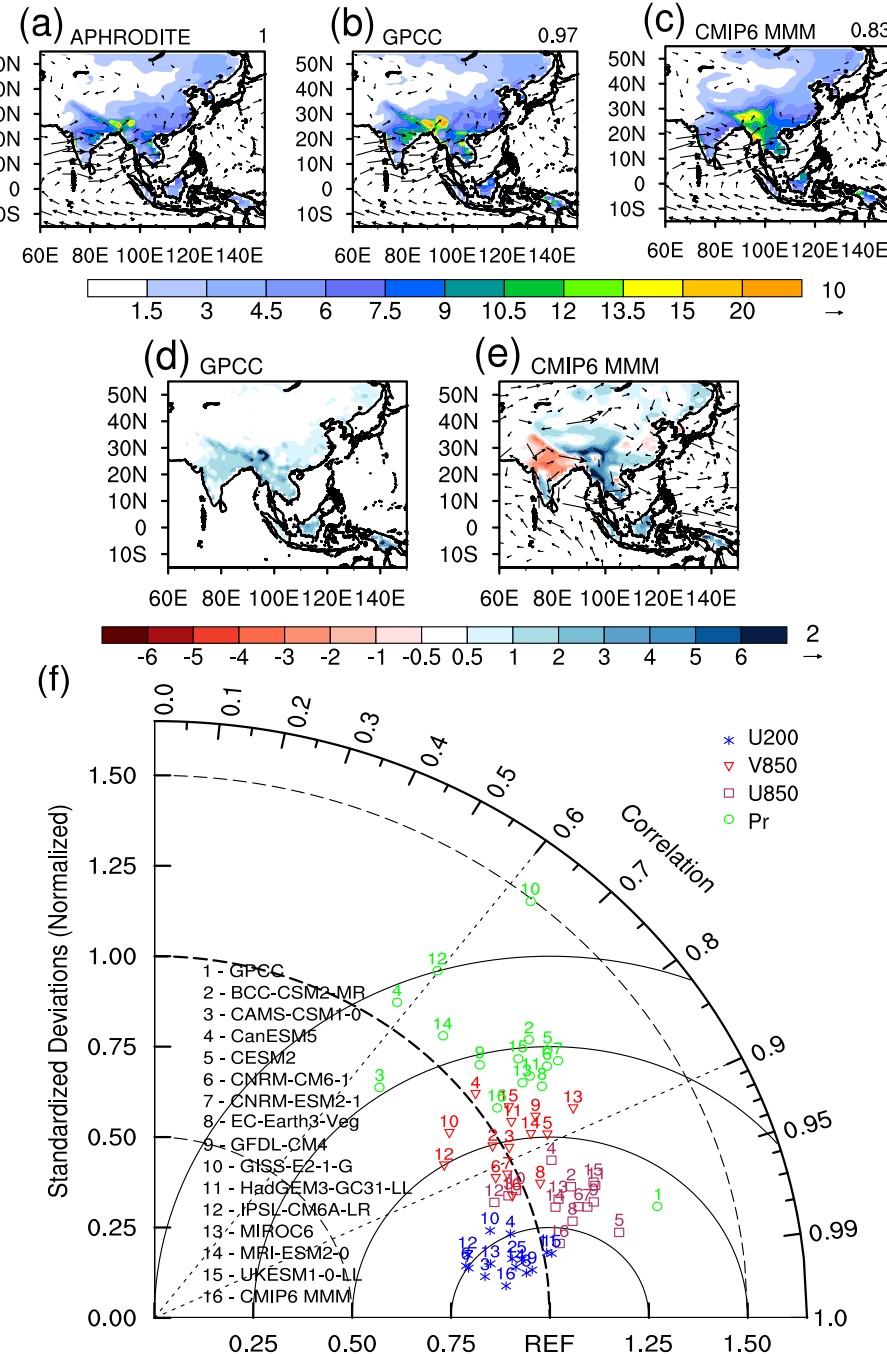

**Figure 5.** JJA-mean 1980-2014 mean precipitation over land [mm day$^{-1}$] overlaid with 850 hPa wind [m s$^{-1}$] from (a): APHRODITE and ERA-Interim; (b): GPCC and ERA-Interim; (c): CMIP6 (multi-model mean). Values in the top right corner of panels (a)-(c) show the pattern correlation with APHRODITE precipitation. (d): Precipitation bias in GPCC relative to APHRODITE; (e): CMIP6 precipitation relative to APHRODITE and CMIP6 850 hPa winds relative to ERA-Interim. (f): Taylor diagram showing the relationship between individual CMIP6 models, the CMIP6 multi-model mean (point 16), GPCC (point 1), and APHRODITE precipitation and ERA-Interim winds.

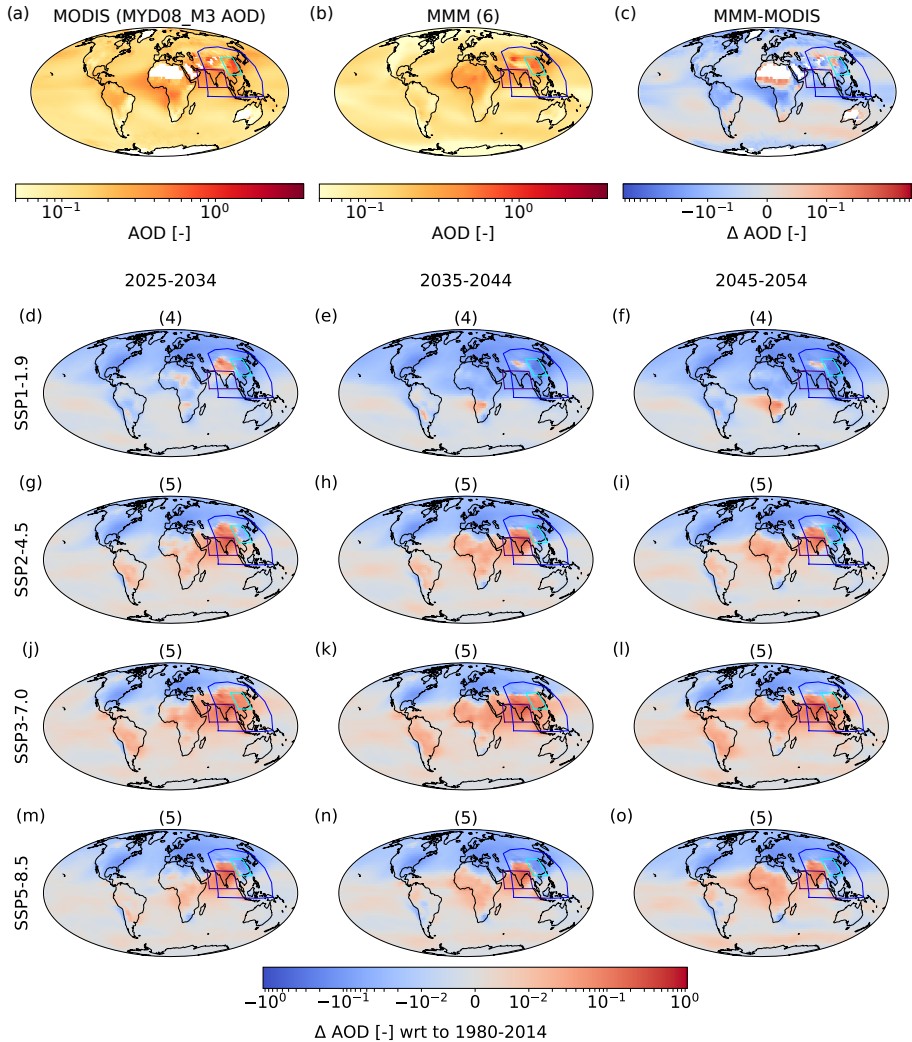

**Figure 6.** (a): 2002-2014 mean aerosol optical depth at 550 nm from MODIS; (b): 2002-2014 mean CMIP6 multi-model mean aerosol optical depth at 550 nm (based on 6 models (Table 1); (c): CMIP6 bias relative to MODIS. CMIP6 AOD anomalies for 2025-2034, 2035-2044, and 2045-2054 vs. 1980-2014 for (d): SSP1-1.9; (e): SSP2-4.5; (f):SSP3-7.0, and (g):SSP5-8.5. For SSP1-1.9 the anomalies are based on 4 models. For all other panels, 5 models are used (Table 1). Blue, purple, and turquoise boxes show the 'Asia', 'South Asia', and 'East Asia' regions used in later analysis. Asia is the region bounded by 5-47.5°N, 67.5-145°E, East Asia is the region bounded by 20-40°N and 100-120°E, and South Asia is the region bounded by 5-25°N and 55-95°E.

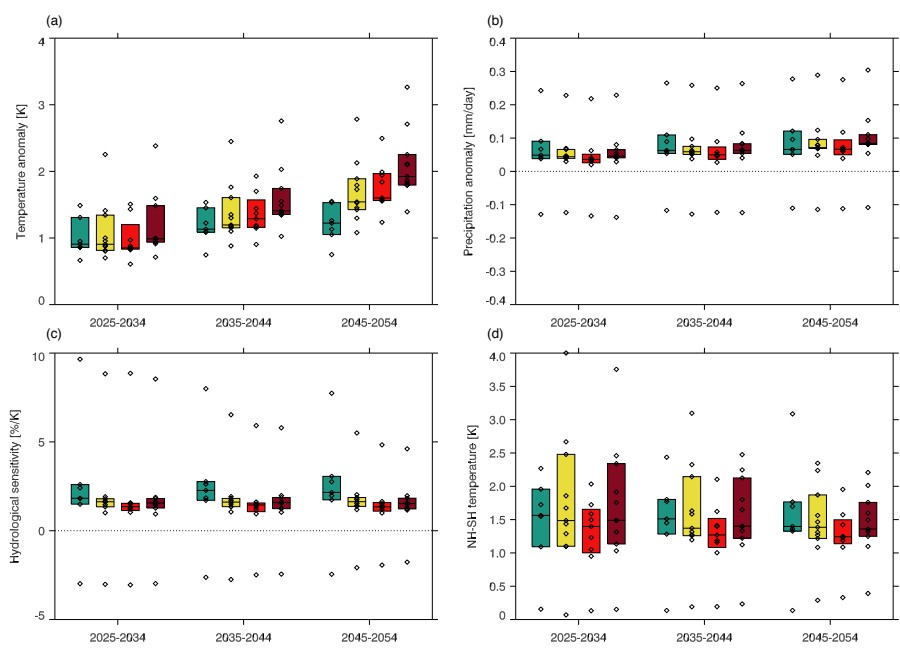

**Figure 7.** Global-mean annual-mean anomalies in (a): near-surface temperature [K]; (b): precipitation [mm day$^{-1}$]; (c): hydrological sensitivity [% K$^{-1}$] relative to 1980-2014. (d): Annual-mean anomalies in interhemispheric temperature gradient [K] relative to 1980-2014. Coloured bars show the interquartile range, and the horizontal bar within this shows the median. Diamonds show values from each model listed in Table 1 as having data availability for a given SSP. Where multiple ensemble members are available the model mean is used.

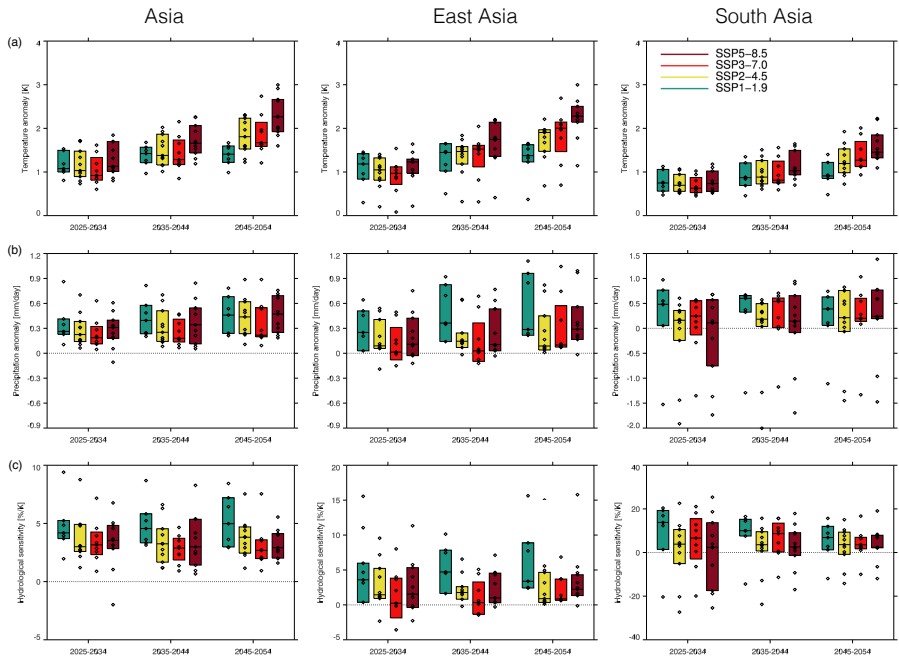

**Figure 8.** Regional mean JJA mean temperature [K], precipitation [mm day$^{-1}$], and hydrological sensitivity [% K$^{-1}$] anomalies relative to 1980-2014 for (a): Asia; (b): East Asia; and (c): South Asia. The three regions are indicated by the boxes in Figure 6. Coloured bars show the interquartile range, and the horizontal bar within this shows the median. Diamonds show values from each model listed in Table 1 as having data availability for a given SSP. Where multiple ensemble members are available the model mean is used. Note that different y-ranges are used for each region in panels (b) and (c).

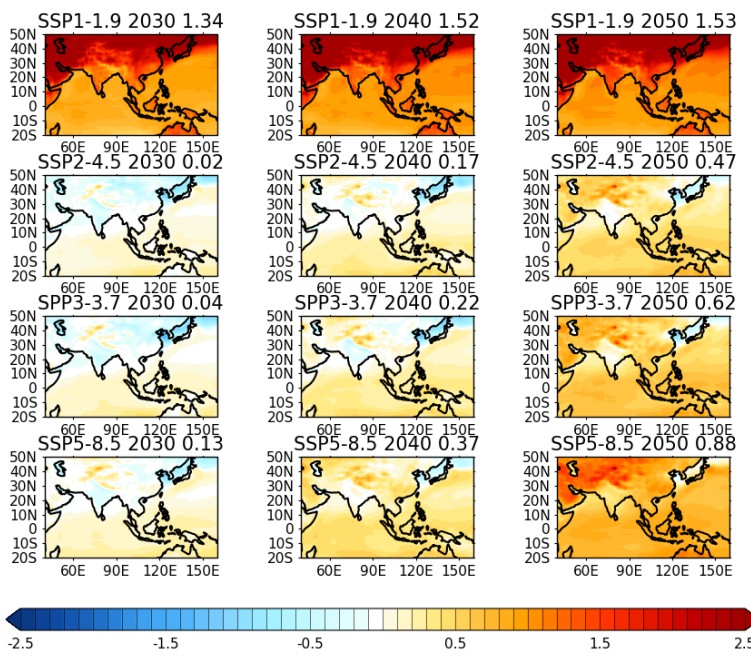

**Figure 9.** (a): CMIP6-mean JJA-mean near-surface temperature anomaly [K] for 2025-2034, 2035-2044, and 2045-2054 vs. 1980-2014 from SSP1-1.9. Relative anomalies for (b): SSP2-4.5.; (c): SSP3-7.0; and (d): SSP5-8.5 vs. SSP1-1.9. The numbers shown at the top right of each panel are the Asian mean, where Asia is the region bounded by 5-47.5°N, 67.5-145°E.

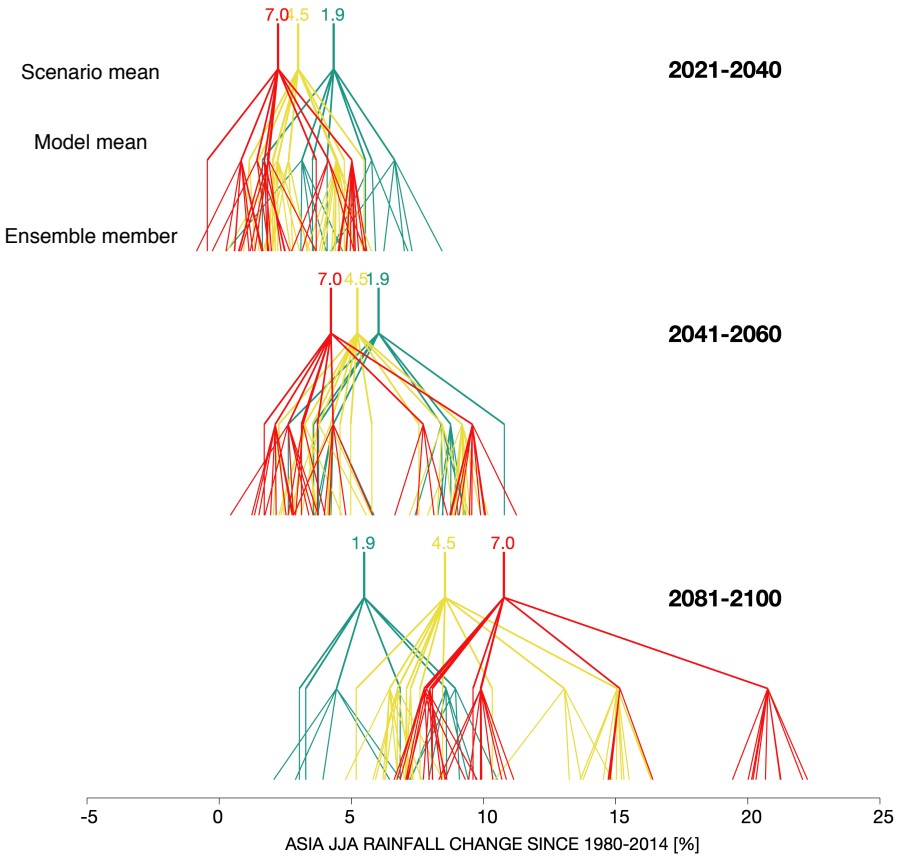

**Figure 10.** Asia-mean JJA-mean precipitation anomaly [%] relative to 1980-2014 in scenario and model means, and individual ensemble members, for 2021-2040, 2041-2060, and 2081-2100, from SSP1-1.9, SSP2-4.5, and SSP3-7.0.

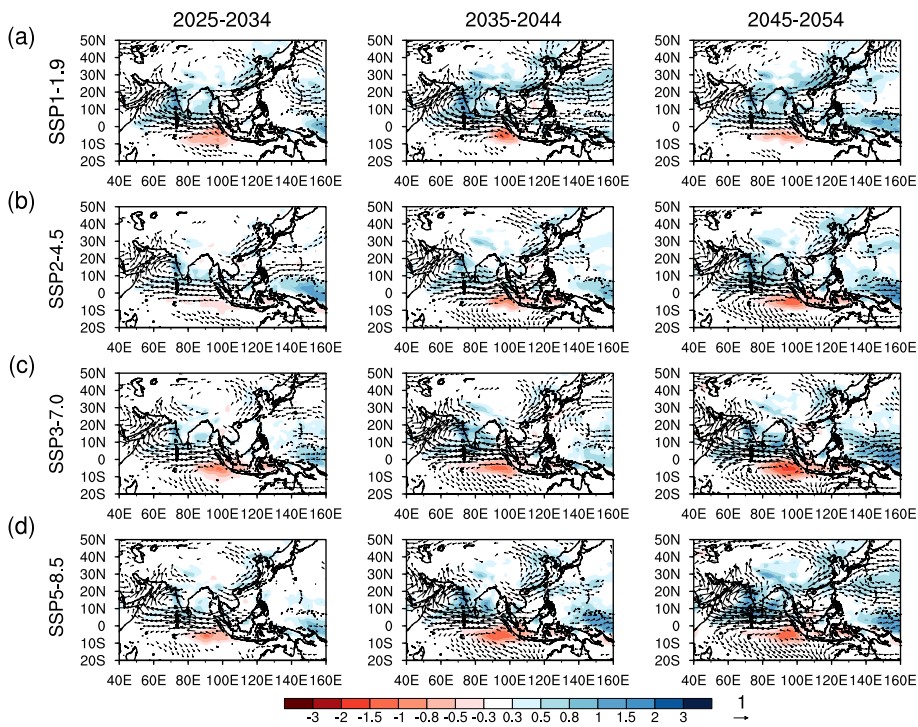

**Figure 11.** CMIP6-mean JJA-mean precipitation [mm day$^{-1}$] and 850 hPa wind anomalies [m s$^{-1}$] for 2025-2034, 2035-2044, and 2045-2054 vs. 1980-2014 from (a): SSP1-1.9; (b): SSP2-4.5; (c): SSP3-7.0; and (d): SSP5-8.5.

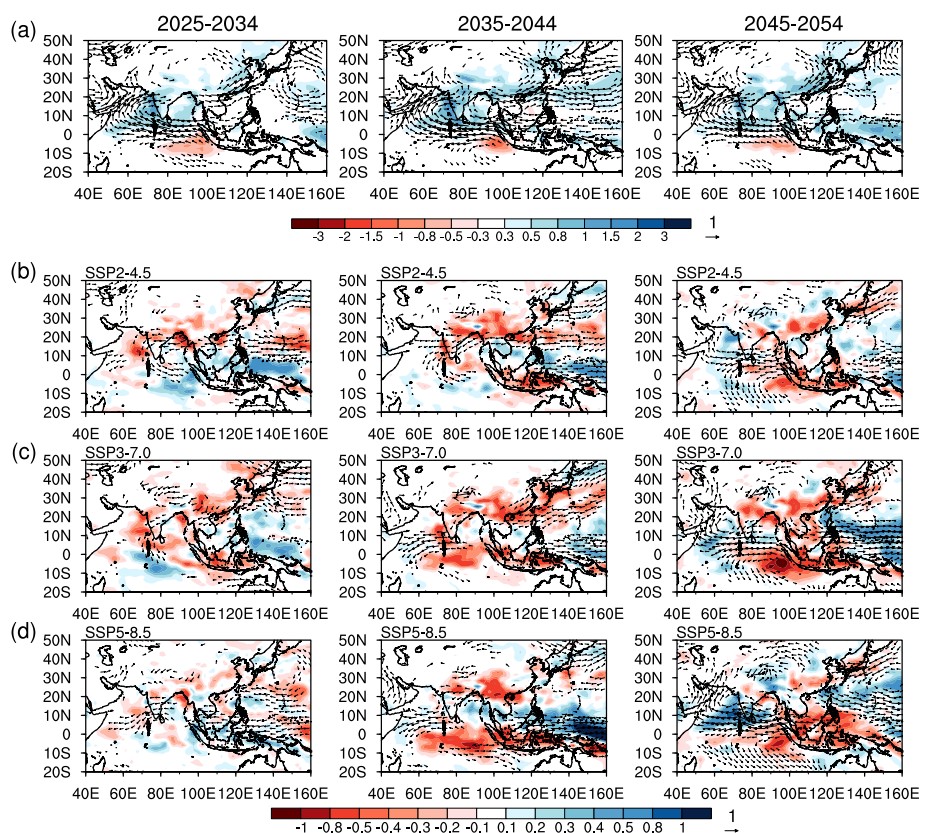

**Figure 12.** (a): JJA-mean precipitation [mm day$^{-1}$] and 850 hPa wind anomalies [m s$^{-1}$] for 2025-2034, 2035-2044, and 2045-2054 vs. 1980-2014 from SSP1-1.9. Anomalies from (b): SSP2-4.5; (c): SSP3-7.0; and (d): SSP5-8.5 relative to the anomalies from SSP1-1.9. To enable a fair comparison of the precipitation patterns, only models with data availability for all scenarios are used (see Table 1).

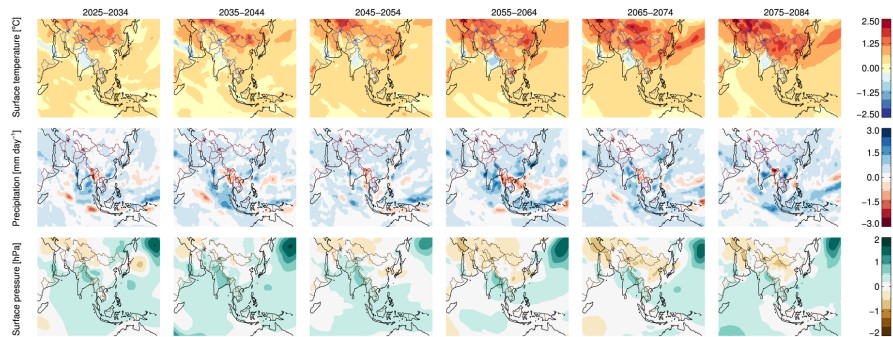

**Figure 13.** JJA-mean (a): near-surface temperature [K]; (b): precipitation [mm day$^{-1}$]; and (c): sea level pressure [hPa] anomalies for 10 year periods vs. 1980-2014 from an anthropogenic aerosol only version of SSP2-4.5 (SSP2-4.5-aer) with MIROC6.

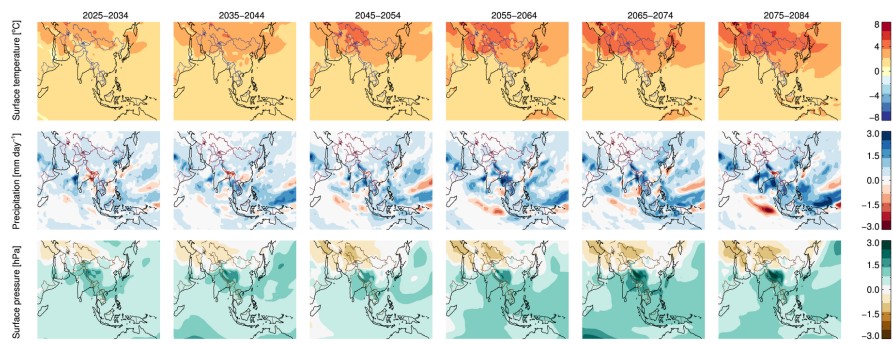

**Figure 14.** JJA-mean (a): near-surface temperature [K]; (b): precipitation [mm day$^{-1}$]; and (c): sea level pressure [hPa] anomalies for 10 year periods vs. 1980-2014 from SSP2-4.5 in MIROC6.

Table 1. CMIP6 models and the number of ensemble members for each used in this work.

| Centre | Model | AOD | | | | | T, P, U850, and V850 | | | | Data reference |
|---|---|---|---|---|---|---|---|---|---|---|---|
| | | hist | 1-1.9 | 2-4.5 | 3-7.0 | 5-8.5 | hist | 1-1.9 | 2-4.5 | 3-7.0 | |
| BCC | BCC-CSM2-MR | | | | | | 3 | 1 | 1 | 1 | Wu et al. (2018); Xin et al. (2019) |
| CAMS | CAMS-CSM1-0 | | 1 | | | | 1 | 1 | 1 | 1 | Rong (2019a); Rong (2019b) |
| CCCma | CanESM5 | 1 | | 1 | 1 | 1 | 10 | 5 | 10 | 10 | Swart et al. (2019b); Swart et al. (2019c) |
| CNRM-CERFACS | CNRM-CM6-1 | 6 | | 6 | 6 | 6 | 10 | 5 | 5 | 6 | Voldoire (2018); Voldoire (2019b) |
| CNRM-CERFACS | CNRM-ESM2-1 | 5 | 5 | 5 | 5 | 5 | 5 | 5 | 5 | 5 | Seferian (2018); Seferian (2019b) |
| EC-Earth-Consortium | EC-Earth3-Veg | | | | | | 1 | 1 | 1 | | (EC-Earth); (EC-Earth) |
| IPSL | IPSL-CM6A-LR | 9 | 1 | 2 | 9 | 1 | 10 | 1 | 2 | 10 | Boucher et al. (2018a); Boucher et al. (20... |
| MIROC | MIROC6 | | | | | | 3 | 1 | 3 | 3 | Tatebe and Watanabe (2018) |
| MOHC | HadGEM3-GC31-LL | | | | | | 4 | | | | Ridley et al. (2019) |
| MOHC | UKESM1-0-LL | 6 | 5 | 5 | 5 | 5 | 4 | 4 | 5 | 5 | Tang et al. (2019); Good et al. (2019) |
| MRI | MRI-ESM2-0 | | | | | | 3 | 1 | 1 | 1 | Yukimoto et al. (2019a); Yukimoto et al. |
| NASA-GISS | GISS-E2-1-G | | | | | | 5 | | | | for Space Studies (NASA/GISS) (2018) |
| NCAR | CESM2 | 10 | | | | | 6 | | | | Danabasoglu (2019a) |
| NOAA-GFDL | GFDL-CM4 | | | | | | 1 | | 1 | | Guo et al. (2018b); Guo et al. (2018a) |

**Table 2.** Historical effective radiative forcing (ERF), calculated from RFMIP sstclim experiments, and ECS from Zelinka et al. (2020). Models shown in *italics* are only used in historical analysis, and do not appear in Figures 7 to 11. Starred(*) models are used in the calculation of the ERF shown in Figure 2.

| Centre | Model | ERF [W m$^{-2}$] | | | ECS | RFMIP data reference |
| | | AA | GHG | LU | | |
| --- | --- | --- | --- | --- | --- | --- |
| CCCma | CanESM5 | | | | 5.64 | |
| CNRM-CERFACS | CNRM-CM6-1 | -1.15 | 2.64 | | 4.90 | Voldoire (2019a) |
| CNRM-CERFACS | CNRM-ESM2-1 | -0.74 | 2.41 | -0.07 | 4.79 | Seferian (2019a) |
| IPSL* | IPSL-CM6A-LR | -0.59 | 2.84 | -0.02 | 4.56 | Boucher et al. (2018b) |
| MIROC* | MIROC6 | -1.06 | 2.69 | -0.03 | 2.60 | Sekiguchi and Shiogama (2019) |
| *MOHC** | *HadGEM3-GC31-LL* | *-1.10* | *3.09* | *-0.11* | *5.55* | *Andrews (2019)* |
| MOHC* | UKESM1-0-LL | -1.11 | 2.97 | -0.18 | 5.36 | O'Connor et al. (2019) |
| MRI* | MRI-ESM2-0 | -1.19 | 3.03 | -0.18 | 3.13 | Yukimoto et al. (2019b) |
| *NASA-GISS** | *GISS-E2-1-G* | *-1.32* | *2.92* | *-0.00* | *2.71* | *for Space Studies (NASA/GISS) (2019)* |
| *NCAR** | *CESM2* | *-1.37* | *3.04* | *-0.04* | *5.15* | *Danabasoglu (2019b)* |
| NOAA-GFDL* | GFDL-CM4 | -0.73 | 3.14 | -0.33 | 3.89 | Paynter et al. (2018) |