# Peer review of "Accelerated increases in global and Asian summer monsoon precipitation from future aerosol reductions"

_Atmospheric Chemistry and Physics, 2019_

## Referee Comment (RC1) · Bryce Harrop (Referee) · 5 Mar 2020

**General comments**

The manuscript makes use of the available CMIP6 SSP projection simulations to evaluate the impact of changing aerosols on the hydrological cycle over South Asia and East Asia. Despite the lack of clean experiments (non-aerosol differences occur across SSPs), the authors argue that simple and robust patterns appear that fingerprint the role of aerosol uncertainty on changes in precipitation, most notably during the first half of the 21st century. It is often difficult, however, to follow the line of reasoning used in the text of the manuscript when examining the figures presented. I have made a note

of several such passages that seem to disagree with what is presented in the figures in the specific comments. There are also several points of discussion in the manuscript relating global scale and regional scale differences, but there is little evaluation presented for which scales are important for which findings. A clearer definition of what constitutes agreement with the hypotheses would make this manuscript much easier to follow. Finally, in addition to discussions about the role of GHGs vs aerosol, there is no mention of land use/land cover change and the impact of its differences between SSPs on rainfall over South Asia or East Asia in this manuscript.

Specific comments

1. The authors argue that, "If the magnitude of the anomaly decreases monotonically from SSP1-1.9, which has the largest aerosol reduction, to SSP3-7.0, which has a moderate aerosol increase, this indicates that aerosol changes are the main driver of the climate response." When looking at the global emissions of BC and SO2 presented in Figure 1, this seems reasonable, but the same logic appears to be applied regionally in this manuscript. Looking at South Asia during the 2015-2050 period, SO2 emissions are highest for SSP5-8.5 and nearly equal for SSP2-4.5 and SSP3-7.0. How are we meant to disentangle the regional and global scale impacts for this region?

2. SSP2-4.5 and SSP5-8.5 are said to have "similar aerosol pathways," and globally that appears to be the case (Fig 1). Again, however, over South Asia, the differences in BC and SO2 emissions between SSP2-4.5 and SSP5-8.5 appear to be as large as their differences relative to SSP3-7.0. This point is raised again in the discussion of Fig 4 where the authors state, "SSP5-8.5 has similar aerosol changes to SSP2-4.5, consistent with the similar changes in emissions (Figure 1)." Given how dissimilar the regional emissions are in Figure 1, it is disconcerting that the AOD pattern for SSP5-8.5 is left off Figure 4, as this would allow readers to accurately see how similar or not the regional emissions are.

3. Figures 5 and 6 show the model mean responses (as points), as well as their

interquartile spread, for global (fig 5) and regional (fig 6) metrics. The temperature responses show noticeable spread between the different pathways, particularly by 2045-2054, but the precipitation responses have far less separation between pathways. I found it difficult to parse what measure the authors use to decide whether precipitation has increased or decreased between pathways. I began by assuming they were referring to the median (which I assume is the horizontal line in each bar). If that were true, then the statement, "Global aerosol reductions in SSP1-1.9 briefly cause this scenario to warm faster than the others considered over Asia and East Asia. . ." should be changed to refer only to East Asia, as Fig 6a (left panel) does not show a larger median temperature anomaly for SSP1-1.9 than SSP2-4.5. Additionally, the statement, "Over Asia, the largest mean precipitation increase occurs, for all decades, in SSP1-1.9. . ." is difficult to parse when it isn't clear if the "mean precipitation" is even marked in the figure. Is the bar actually the multi-model mean? If that is true, then the increase in precipitation over Asia is larger in both SSP2-4.5 and SSP5-8.5 than it is in SSP1-1.9. These two figures, and their accompanying text, must be clarified before any rigorous evaluation of the conclusions can be made. I also strongly recommend adding some discussion of when differences between regional precipitation changes at the decadal scale are statistically significant, or at a minimum robust across models.

4. The cooling over India is argued as the reason for suppressed precipitation increases in SSP2-4.5 and SSP5-8.5 relative to SSP1-1.9 and SSP3-7.0, but the cooling in Figure 7 is strongest for SSP3-7.0. How does one reconcile this? On a similar note, why are the temperature anomalies for South Asia and East Asia all positive in Figure 6a when Figure 7 shows cooling for SSP2-4.5, SSP3-7.0, and SSP5-8.5 for 2025-2034?

5. The warming and rainfall change patterns for the two individual SSP2-4.5-aer simulations are difficult to compare to the multimodel mean, and even to the rainfall response in Figure S8 (owing to changes in both the range of the colorbar and the colors used). It would be useful to show a direct comparison of the full SSP2-4.5 response to

that of SSP2-4.5-aer for each of the two models available so that an assessment can be made for how much the climate responses are indeed driven by aerosols.

6. Figures are too small to be readable when printed, and the quality is so low that they are hard to read even when zoomed in on a computer. Please consider revising with vector graphics or higher DPI raster images. It would be helpful to readers to add an outline of the analysis regions (Asia, S. Asia, and E. Asia) to the map plots. Please maintain a consistent map projection for all map plots. Please also be consistent with colorscales so that metrics can be compared across figures (e.g., Fig 7 vs Fig 11, or Fig 9 vs Fig 11). Finally, please consider changing Fig 4c to be MMM-MODIS so that it is consistent with the caption.

Technical corrections

Page 2, line 34, "AA" is not defined Page 4, line 7 typo "has yet to be emerge" Page 6, line 6 typo "present - day" Figure 2 caption typo "180-2014" Figure 7, there is a change in font between subpanels

---

## Referee Comment (RC2) · Anonymous Referee #2 · 9 Mar 2020

This study investigates the possible influences of different aerosol reductions in the future on the global and Asia surface temperature and rainfall. The topic is quite important, but the method they took may have some problems, at least for some conclusions. Their writing is very unclear (with many typos, which greatly affect the reading experience) and very hard to follow. At the same time, the figures are so small and so unclear (also the captions) that I try my best to understand what they showed. Besides these, I still have several major comments and I don't think this manuscript can be accepted unless all these concerns are well addressed.

Major comments: 1. Due to the lack of clean experiments, the guidance to distinguish

the relative importance of GHG and aerosol forcing in this study is that different scenarios may be similar in one forcing change, while very different in the other forcing change. This seems plausible, but the question is whether the other forcings (e.g., land use) keep unchanged in different scenarios. I guess probably not. So the question is whether they are important or not for the main conclusion drawn here. I think the authors should seriously think about it and do some analysis on it. 2. From Fig. 5, it seems that for the global mean precipitation and hydrological sensitivity, the responses of most models are close to each other, except two models with totally opposite signs (one with large positive value and the other with large negative value). Could you do more analysis on these two models? With the same aerosol emission, how can these two models produce totally opposite results? To me, I know the aerosol forcing has large uncertainty (should affect the results in a quantitative way), but in a qualitative way, it should be the same result at least at the global mean. Hence, it quite surprises me. In Fig. 6, it seems that over Asia, the uncertainty is smaller, at least not opposite.

Specific comments: 1. Page2L35: Why is this case? It is hard to understand. It is better to provide an explanation here. 2. Page3L13: full->fully 3. Page3L14: add a period. 4. Page7L3-4: You should clearly state this in the figure caption to make sure each figure can be understood from the figure itself. 5. Page7L18: Please add "partly". I don't think aerosol forcing explains all the weakening of Asian summer monsoon. 6. Page7L30: remove "the" 7. Section 4.1: I don't think it is suitable to compare the SSP2-4.5-aer simulations from two models with SSP2-4.5 simulations from all models. You should compare these two simulations from the same model.

---

## Author Comment (AC1) · 15 Jun 2020

We thank the reviewers for their constructive comments. In our response, referee comments are indicated in **bold**, with our comments and changes to the manuscript in plain text. In addressing the reviewers' comments, we have added a new figure to the manuscript. Throughout our response, when discussing figures, we give both the original and revised figure number.

**Reviewer 1: Bryce Harrop**

[Figure]

**The manuscript makes use of the available CMIP6 SSP projection simulations to evaluate the impact of changing aerosols on the hydrological cycle over South Asia and East Asia. Despite the lack of clean experiments (non-aerosol differences occur across SSPs), the authors argue that simple and robust patterns appear that fingerprint the role of aerosol uncertainty on changes in precipitation, most notably during the first half of the 21st century. It is often difficult, however, to follow the line of reasoning used in the text of the manuscript when examining the figures presented. I have made a note of several such passages that seem to disagree with what is presented in the figures in the specific comments. There are also several points of discussion in the manuscript relating global scale and regional scale differences, but there is little evaluation presented for which scales are important for which findings. A clearer definition of what constitutes agreement with the hypotheses would make this manuscript much easier to follow. Finally, in addition to discussions about the role of GHGs vs aerosol, there is no mention of land use/land cover change and the impact of its differences between SSPs on rainfall over South Asia or East Asia in this manuscript.**

Thank you Bryce for the thoughtful and detailed review. We have added detail to the text throughout the manuscript, which hopefully makes our reasoning clearer. Where you had specific concerns about particular paragraphs, we have addressed them in the manuscript and respond to them directly below.

AR5 suggested that land use forcing was an order of magnitude smaller than that from anthropogenic aerosols, so we didn't consider it in the original manuscript. However, we have now looked into the details of the experiments in CMIP6, and the available literature, and agree that it is important to mention this. We have now included a summary of land use changes in our initial description of the SSPs, and commentary on their potential role in the manuscript.

Where data are available, we have calculated the global mean ERF due to anthropogenic aerosol changes and to land use changes. We have included these values in Table 2, alongside the ERF from greenhouse gas increases, and the Equilibrium Climate Sensitivity. The forcing from land use is much smaller than that due to aerosol. However, we note in the manuscript that it may be of more importance locally.

**Specific comments**

**1. The authors argue that, "If the magnitude of the anomaly decreases monotonically from SSP1-1.9, which has the largest aerosol reduction, to SSP3-7.0, which has a moderate aerosol increase, this indicates that aerosol changes are the main driver of the climate response." When looking at the global emissions of BC and SO2 presented in Figure 1, this seems reasonable, but the same logic appears to be applied regionally in this manuscript. Looking at South Asia during the 2015-2050 period, SO2 emissions are highest for SSP5-8.5 and nearly equal for SSP2-4.5 and SSP3-7.0. How are we meant to disentangle the regional and global scale impacts for this region?**

Disentangling regional and global scale impacts is a study in itself, and an interesting one. It wouldn't be possible to do with the type of experiments that we consider here. There are a number of published studies that look at the relative roles of local and remote aerosol emissions for monsoon changes. We now refer to these in the manuscript, and make clear that when we look at the monsoon response in the SSPs we are considering the effect of both local and remote aerosol changes.

**2. SSP2-4.5 and SSP5-8.5 are said to have "similar aerosol pathways," and globally that appears to be the case (Fig 1). Again, however, over South Asia, the differences in BC and SO2 emissions between SSP2-4.5 and SSP5-8.5 appear to be as large as their differences relative to SSP3-7.0. This point is raised again in the discussion of Fig 4 where the authors state, "SSP5-8.5 has similar aerosol**

**changes to SSP2-4.5, consistent with the similar changes in emissions (Figure 1).” Given how dissimilar the regional emissions are in Figure 1, it is disconcerting that the AOD pattern for SSP5-8.5 is left off Figure 4, as this would allow readers to accurately see how similar or not the regional emissions are.**

We have added the AOD for SSP5-8.5 to Figure 4, and more clearly delineated our discussion of regional and global aerosol when introducing the SSPs.

In our discussion of the results we now refer to the different characteristics of the emission pathways over South Asia compared to the global and East Asian case, and discuss the impact of this in the context of the monsoon changes.

**3. Figures 5 and 6 show the model mean responses (as points), as well as their interquartile spread, for global (fig 5) and regional (fig 6) metrics. The temperature responses show noticeable spread between the different pathways, particularly by 2045-2054, but the precipitation responses have far less separation between pathways. I found it difficult to parse what measure the authors use to decide whether precipitation has increased or decreased between pathways. I began by assuming they were referring to the median (which I assume is the horizontal line in each bar). If that were true, then the statement, “Global aerosol reductions in SSP1-1.9 briefly cause this scenario to warm faster than the others considered over Asia and East Asia...” should be changed to refer only to East Asia, as Fig 6a (left panel) does not show a larger median temperature anomaly for SSP1-1.9 than SSP2-4.5. Additionally, the statement, “Over Asia, the largest mean precipitation increase occurs, for all decades, in SSP1-1.9...”is difficult to parse when it isn’t clear if the “mean precipitation” is even marked in the figure. Is the bar actually the multi-model mean? If that is true, then the increase in precipitation over Asia is larger in both SSP2-4.5 and SSP5-8.5 than it is in SSP1-1.9. These two figures, and their accompanying text, must be clarified before any rigorous evaluation of the conclusions can be made. I also strongly**

**recommend adding some discussion of when differences between regional pre-cipitation changes at the decadal scale are statistically significant, or at a mini-mum robust across models.**

We have now included a paragraph clarifying the approach used in Figures 5 and 6 (revised Figures 6 and 7). The horizontal bars are the median, and we have now taken care to refer to this consistently in the text, rather than referring to the mean. We now include a discussion of significance and robustness throughout this section. For our sample size, the 95% confidence interval about the median is typically very close to the interquartile range, based on the empirical relation in McGill et al. (1978). To account for the asymmetry in the distribution of models about the median in some cases, we use the interquartile range to determine significance.

**4. The cooling over India is argued as the reason for suppressed precipitation in-creases in SSP2-4.5 and SSP5-8.5 relative to SSP1-1.9 and SSP3-7.0, but the cooling in Figure 7 is strongest for SSP3-7.0. How does one reconcile this? On a similar note, why are the temperature anomalies for South Asia and East Asia all positive in Figure 6a when Figure 7 shows cooling for SSP2-4.5, SSP3-7.0, and SSP5-8.5 for2025-2034?**

Figure 6 (revised Figure 7) shows an anomaly relative to 1980-2014, so includes a considerable amount of global warming. Figure 7 (revised Figure 8) shows the same for SSP1-1.9. For the other scenarios in Figure 7 (revised Figure 8), we show a difference relative to SSP1-1.9 to try to highlight the differences between the scenarios. This is the reason for the apparent change in sign between Figures 6 and 7 (revised Figures 7 and 8), and we have clarified this in the text and the caption.

We have removed the argument for cooling as the reason for suppressed precipitation since precipitation changes can also lead to temperature changes.

**5. The warming and rainfall change patterns for the two individual SSP2-4.5-aer simulations are difficult to compare to the multimodel mean, and even to the rainfall response in Figure S8 (owing to changes in both the range of the colorbar and the colors used). It would be useful to show a direct comparison of the full SSP2-4.5 response to that of SSP2-4.5-aer for each of the two models available so that an assessment can be made for how much the climate responses are indeed driven by aerosols.**

This comparison is now included. We show both SSP2-4.5-aer and SSP2-4.5 for MIROC6 in the main text (revised Figures 12 and 13), and SSP2-4.5-aer and SSP2-4.5 for CanESM5 in the supporting information (Supplementary Figures 7 and 8). We now use consistent colours for our precipitation scales throughout the manuscript to facilitate comparison between figures.

**6. Figures are too small to be readable when printed, and the quality is so low that they are hard to read even when zoomed in on a computer. Please consider revising with vector graphics or higher DPI raster images. It would be helpful to readers to add an outline of the analysis regions (Asia, S. Asia, and E. Asia) to the map plots. Please maintain a consistent map projection for all map plots. Please also be consistent with colorscales so that metrics can be compared across figures (e.g., Fig 7 vs Fig 11, or Fig 9 vs Fig 11). Finally, please consider changing Fig 4c to be MMM-MODIS so that it is consistent with the caption.**

We have provided both vector and higher DPI raster images to ACP, and added outlines of the analysis regions to Figure 4.

All regional plots now use the same domain, except for Figure 3 (revised Figure 4), S1, and S2, where we use a slightly smaller domain. These figures show a comparison to APHRODITE, which has a limited data domain.

The different magnitudes in Figures 7-11 (revised manuscript: Figures 8-12) made

it difficult to use exactly the same colour scale throughout. However, we have now standardised the type of colour scale used for each variable, so that temperature is now blue:yellow:red, precipitation is red:white:blue, and sea level pressure is brown:white:green throughout.

We have made the suggested change to Figure 4c (revised Figure 5c).

**Technical corrections**

**Page 2, line 34, "AA" is not defined  Page 4, line 7 typo "has yet to be emerge" Page 6,line 6 typo "present - day"  Figure 2 caption typo "180-2014"  Figure 7, there is a change in font between subpanels**

All now corrected, thank you.

**References**

Robert McGill, John W. Tukey and Wayne A. Larsen. Variations of Box Plots, The American Statistician, Vol. 32, No. 1 (Feb., 1978), pp. 12-16

**Reviewer 2**

**This study investigates the possible influences of different aerosol reductions in the future on the global and Asia surface temperature and rainfall. The topic is quite important, but the method they took may have some problems, at least for some conclusions. Their writing is very unclear (with many typos, which greatly affect the reading experience) and very hard to follow. At the same time, the figures are so small and so unclear (also the captions) that I try my best to understand what they showed. Besides these, I still have several major comments**

**and I don't think this manuscript can be accepted unless all these concerns are well addressed.**

We thank the reviewer for their comments and are sorry to hear that they found our writing unclear. We have corrected the typos identified by both reviewers, made changes to the text to further improve the clarity. We have added more detail about our methodology. We have also added extra detail to the captions of Figures 5, 6, 7, 8, 9, and 10, and included either higher resolution or vector versions of all figures. We have also addressed the reviewer's detailed comments in the manuscript, and provide responses for those separately below.

**Major comments:**

**1. Due to the lack of clean experiments, the guidance to distinguish the relative importance of GHG and aerosol forcing in this study is that different scenarios may be similar in one forcing change, while very different in the other forcing change. This seems plausible, but the question is whether the other forcings (e.g.,land use) keep unchanged in different scenarios. I guess probably not. So the question is whether they are important or not for the main conclusion drawn here. I think the authors should seriously think about it and do some analysis on it.**

The SSPs do include a range of land use changes in addition to a range of aerosol pathways. We have now included a summary of land use changes in our initial description of the SSPs, and commentary on their potential role in our results. There is a limited amount of literature available that already compares the relative roles of anthropogenic aerosol and land use changes in monsoon changes, and we now refer to this in the text. This work suggests that the response to anthropogenic aerosol changes is larger than the response to land use changes over China, but that land use changes may be important over India.

[Figure]

[Figure]

Where data are available, we have calculated the global mean effective radiative forcing (ERF) due to anthropogenic aerosol changes and to land use changes. We have added these values to the manuscript in Table 2, alongside the ERF from greenhouse gas increases, and the Equilibrium Climate Sensitivity. The forcing from land use is much smaller than that due to anthropogenic aerosol. However, as we now note in the discussion, it may be of more importance locally. Overall, it looks like the land use changes will drive monsoon changes of the same sign as the aerosol driven changes, and we have also noted this in the manuscript.

Given that the forcing from land use changes are so small compared to the forcing from anthropogenic aerosol, we think it would be distracting to include analysis beyond a comparison of the radiative forcings and a discussion of the relevant literature in this paper.

**2. From Fig. 5, it seems that for the global mean precipitation and hydrological sensitivity, the responses of most models are close to each other, except two models with totally opposite signs (one with large positive value and the other with large negative value). Could you do more analysis on these two models? With the same aerosol emission, how can these two models produce totally opposite results? To me, I know the aerosol forcing has large uncertainty (should affect the results in a quantitative way), but in a qualitative way, it should be the same result at least at the global mean. Hence, it quite surprises me. In Fig. 6, it seems that over Asia, the uncertainty is smaller, at least not opposite.**

The outlying models in Figure 5, and the large opposite responses from two models in Figure 5c, are mainly the result of our choice to show anomalies relative to 1980-2014, rather than large differences in absolute values across the models. These anomalies for each SSP include a large amount of global warming, and the difference between the outlying models is largely a reflection of different climate sensitivities, rather than differences in the response to aerosol forcing. For each scenario, the outlying models

are the same in each case, so have no influence on the relative differences between the scenarios.

Figure 1 of this response shows the temperature time series that are used in Figure 5 (revised Figure 6). SSP2-4.5 is used as an example. Panel (a) shows the absolute values of global-mean JJA-mean near-surface temperature. The outlying models from Figure 5 (revised Figure 6) are highlighted with bolder lines. Panel (b) shows the same data as anomalies relative to 1980-2014, which is what we show in the paper. Comparison of the two panels demonstrates that the models are not unusual in their mean climate, or the sign of the trend, but do warm relatively more (or less) than the other models between 1980 and 2020.

We have done some further analysis of the outlying models from Figure 5 (revised Figure 6), as suggested by the reviewer. Globally, the low outliers are MIROC6 (temperature) and CAMS-CSM1-0 (precipitation), while the high outliers are EC-Earth-Veg, UKESM, and CanESM5 (temperature) and UKESM (precipitation). These models are those with the lowest and highest climate sensitivities in our subset, consistent with them having the smallest and largest trends over 1980-2014 (as shown in Figure **??** of this response). These points are now noted in the manuscript. Maps of the precipitation responses in the individual models are shown in Figure S10.

**Specific comments:**

**1. Page2L35: Why is this case? It is hard to understand. It is better to provide an explanation here.** We now explicitly state that future warming is driven by a combination of positive radiative forcing from greenhouse gas increases and positive radiative forcing from anthropogenic aerosol decreases, so that a weaker aerosol forcing results in a more moderate warming.

**2. Page3L13: full->fully** Done

**3. Page3L14: add aperiod.** Done

**4. Page7L3-4: You should clearly state this in the figure caption to make sure each figure can be understood from the figure itself.** Details of the quantities shown in the box plots have been added to the captions for Figures 5 and 6 (revised Figures 6 and 7).

**6. Page7L18: Please add "partly". I don't think aerosol forcing explains all the weakening of Asian summer monsoon.** Changed to 'largely'. We accept that a single forcing is unlikely to explain all of the weakening, but there is good evidence that aerosol forcing is the dominant driver (relevant papers cited in manuscript).

**6.Page7L30: remove "the"** This sentence has been rewritten.

**7. Section 4.1: I don't think it is suitable to compare the SSP2-4.5-aer simulations from two models with SSP2-4.5 simulations from all models. You should compare these two simulations from the same model.** This comparison is now included. We show both SSP2-4.5-aer and SSP2-4.5 for MIROC6 in the main text (Figures 12 and 13), and SSP2-4.5-aer and SSP2-4.5 for CanESM5 in the supporting information (Figures S7 and S8).

**Additional changes not requested by the reviewers**

There was a problem with the secondary organic aerosol in the CESM SSPs and the data has been withdrawn: https://errata.es-doc.org/static/view.html?uid=eb69632c-a6e2-7667-a112-a98b7745e2ea We have removed these simulations from our analy-
sis.

In the submitted version of the manuscript there were data points with a temperature anomaly of 0K in Figure 5a. These were erroneous, and have been corrected in the revised version (Figure 6a).

As part of our attempt to improve the readability of the manuscript, we have replaced the JJA mean interhemispheric temperature gradient originally shown in Figures 2b and 5d with the annual mean, making it consistent with the other panels in the figure. We had originally included JJA here to give a closer link to the monsoon results discussed later in the manuscript. However, the pattern of the response across the SSPs is similar in both seasons, and the use of the annual mean for this panel means that all discussion in Section 3 is for the same season. The panels from Figure 5 (revised Figure 6) for the annual mean (a) and JJA mean (b) are shown in Figure 2 of this response. There is no qualitative difference between them when comparing the relative position of the median across SSPs.
* * *
[Figure]

**Fig. 1.** (a): Annual-mean global-mean temperature time series for the historical simulation and SSP2-4.5. (b): The same data as shown in panel (a), but presented as an anomaly relative to 1980-2014.

[Figure]

**Fig. 2.** (a): Annual-mean interhemispheric temperature gradient anomalies relative to 1980-2014 from SSP1-1.9, 2-4.5, 3-7.0, and 5-8.5 (as shown in the revised manuscript). (b): As for panel (a), but for JJA

---

## Author Response (AR2)

We thank the reviewers for their comments. In our response, referee comments are indicated in **bold**, with our comments and changes to the manuscript in plain text.

**Reviewer 1**

**First, my thanks to the authors for the responses to my previous comments. While the manuscript has been improved, I feel some of the concerns I had with the initial submission have not been fully addressed. I recommend an additional round of revisions. My first point relates to Figures 2, 6, and 7. I appreciate the addition of Figure 2 and the added description for revised figures 6 & 7. The authors indicate that the median is being used to determine significance, yet the text still refers to the mean response in several places when drawing conclusions about the results.**

We regret that we did not edit all occurrences of 'mean' in the previous revision. All occasions of incorrect use have now been fixed. However, because we are dealing with spatial averages, it is not the case that every reference to a box plot should relate to the 'median'.

**"The influence of aerosol is more clearly seen in regional mean precipitation that regional mean temperature (Figure 7a, b; Figure 2)…"**

This quote is from page 9, line 4 of the previous version of the manuscript. Here, we are referring to the mean temperature and precipitation over the various Asian regions. 'Mean' is correct here, as we are referring to the figure generally, which shows box plots of regional mean values.

**"Over Asia, the largest mean precipitation increase relative to 1980-2014 occurs in SSP1-1.9…"**

This quote is from page 9, line 5 of the previous version of the manuscript. We are indeed referring to the median in this case, and have changed the text accordingly.

**"Over East Asia, JJA mean precipitation is not significantly larger in 1980-2014 in SSP3-7.0 until 2045-2054."**

This quote is from page 9, line 13 of the previous version of the manuscript. We think that the use of 'mean' is correct here. We are referring to the regional mean precipitation. The model median is used to determine the significance, but the discussion still relates to the regional mean precipitation.

**Are the mean responses always the same as the median responses? If so, why not just say the median? If they differ, do they suggest conflicting conclusions?**

All box plots show anomalies in regional mean values. The median shown in these figures is the model median estimate of this regional mean value. Thus, it is correct to refer to the quantities shown in these figures as regional mean temperature and precipitation anomalies at most points in the text. When talking about significant differences between the anomalies from different SSPs it is then necessary to discuss the model median estimate of the regional mean value. We have rephrased some of the sentences in question to clarify whether we are talking about the regional mean of a variable, or the model median estimate of that value.

**Additionally, regarding the statistical significance of differences, Figure 2 suggests the authors determine differences to be significant when the median of one scenario sits beyond the IQR of another scenario. What is done in the case where the median of scenario A is beyond the IQR of scenario B, but the median of scenario B is within the IQR of scenario A, are they considered significantly different? This happens numerous times in Figure 7.**

We consider these differences to be significantly different. Since we have a physical expectation of the sign of the difference, such a one-sided test is appropriate.

**At other times, the responses in Figure 7 fail to meet the significance threshold of Figure 2, but the authors continue to describe the results as if they were significant. I have noted a few examples below.**

All the differences described in the manuscript as significant are correctly identified, where significance is defined as the median for one scenario lying outside the interquartile range of another. We appreciate that this can be difficult to see by eye, so, for the avoidance of doubt, we have provided the relevant numerical values below for all the differences questioned by the reviewer. However, some of the issues raised here are due to our presentation of the schematic in Figure 2 (now Figure 3), so we will first address the reviewer's overall comment, before coming back to the specifics.

**In short, perhaps a different test is needed to support the statements the authors are making. In the manuscript's present form, the conclusions are consistent with the results, but are not supported by them with sufficient rigor. While the conclusions may ultimately be true, it is hard not to remain skeptical that the aerosol signal is "clear" or can be considered the "main driver" of the precipitation differences found in the simulations.**

5 We are considering short time horizons in this work, and as such have no expectation that the anomalies in each SSP will be statistically different to the others. However, there is a physical expectation of a consistent pattern of differences between the SSPs based on the differences in forcing, which may result in significant differences between the most extreme scenarios (SSP1-1.9 and SSP3-7.0 for aerosol, and SSP1-1.9 and SSP5-8.5 for GHGs). Given the physical expectation for a larger decrease in aerosol emissions to produce larger temperature and precipitation increases, and for a larger increase in

10 greenhouse gas emissions to produce larger temperature and precipitation increases (e.g. Li et al., 2015, Lau and Kim, 2017), we consider the qualitative differences between SSPs illustrated in Figure 2 (now Figure 3) to demonstrate an aerosol-driven or greenhouse-gas-driven signal when they are apparent in the data. This can be further supported by statistically significant differences between the extreme scenarios in each case, without the need for each individual SSP to be significantly different to the other.

15 We admit that it was short-sighted to show statistically significant differences between all SSPs in all time periods in Figure 2 (now Figure 3), and can see in the comments that this is the basis for the reviewer's objections to several statements in the manuscript. Figure 2 (now Figure 3) was an attempt at a schematic that would work at a glance, and we're afraid we failed to consider the possibility that it would be used to aid quantitative interpretation of the subsequent figures. In reality it doesn't reflect our expectations of the numerical result. We apologise for the confusion, and have amended the figure so that

20 it now shows both our expectations of the relative positions of the anomalies from each SSP, and the potential for significant differences across the SSPs.

We do not think it would be appropriate to try a number of different statistical tests until we found the one that supported the statements made in the text. We made a deliberate decision not to include indicators of statistical significance in our original submission. The qualitative patterns across the SSPs are in themselves a powerful indicator of the physical climate response

25 to forcing, and we place more weight on a physically consistent pattern than whether or not a given value exceeds what is essentially an arbitrary threshold. In ScenarioMIP, we are dealing with several models with very different aerosol process representations, and temperature and precipitation climatologies. The fact that these models combine to show consistently, over different regions and time periods, a pattern across the SSPs that is not expected from the greenhouse gas radiative forcing, but is consistent with the differences in aerosol emission pathways, is much more compelling than a statistical difference. There

30 is extensive literature that supports aerosol as the main driver of historical trends in the monsoon (a sample of which we now include in the introduction, in addition to our original citations in Section 4), which further supports the physical mechanism underlying these patterns. We agree with the reviewer that a statistical test can be used to give weight to a conclusion, and have now noted significant differences where they arise, but consider this to be of secondary importance compared to consistency with known physical mechanisms and robustness across models.

35 **"There is a clear aerosol-driven signal in future increases in global mean precipitation and hydrological sensitivity" I agree with the statement with respect to hydrologic sensitivity, but the precipitation anomalies are largely statistically indistinguishable from one another. For the NH-SH gradient, there is little indication until maybe 2045-2054, that the aerosols play a significant role there either (as determined using the method outlined in Figure 2) despite the authors claiming, "…anthropogenic aerosol is the main driver of trends in the interhemispheric temperature gradient until**

40 **2050…"**

For global precipitation, relating to Figure 7b, the median values and interquartile ranges match the qualitative 'aerosol-driven' pattern described in Figure 2 (now Figure 3), with SSP1.1-9>SSP2-4.5>SSP3-7.0, and SSP2-4.5 comparable to SSP5-8.5, for 2025-2034 and 2035-2044. The pattern begins to break down in 2045-2054 when SP1-1.9 is no longer greater than SSP3-7.0, and SSP5-8.5 can be seen rising relative to SSP2-4.5. These features underpin our assertion that there is an aerosol

45 driven signal in global mean precipitation in the earlier periods, with more evidence of a GHG-driven response emerging in the later period. We are basing our statements on the qualitative pattern, which demonstrates a physical relationship. This statement is further supported by the fact that the median anomaly in global mean precipitation is significantly less in SSP3-7.0 than in SSP1-1.9 for both 2025-2034 and 2035-2044.

2025-2034: SSP1-1.9 25th percentile = 0.0381300, SSP3-7.0 median = 0.0358596
2035-2044: SSP1-1.9 25th percentile = 0.0536375, SSP3-7.0 median = 0.0498199

**"Global aerosol reductions in SSP1-1.9 briefly cause faster warming over all Asian regions than the other scenarios considered, but this effect does not persist beyond the 2040s (Figure 7a)." Again, there is no statistically significant increase in temperature for SSP1-1.9 relative to the other pathways (except maybe relative to SSP3-7.0 over East Asia during 2025-2034). After continuing into the manuscript, the combination of Figures 5 and 8 seems to lend support for the argument of faster warming in SSP1-1.9, so maybe the manuscript.**

In all three regions, the temperature anomalies in 2025-2034 are ordered across the SSPs according to aerosol emission changes, and match the 'aerosol-driven' pattern shown in Figure 2 (now Figure 3) during this period. For the Asia region, the SSP3-7.0 anomaly is significantly smaller than the SSP1-1.9 anomaly.

2025-2034, Asia: SSP1-1.9 25th percentile = 1.00046, SSP3-7.0 median = 0.918823

As noted in the manuscript, the aerosol signature is very short-lived in the regional temperature anomalies, and they have a GHG-driven pattern in 2045-2054 (with the exception of SSP3-7.0 over Asia), and can be seen transitioning between the two patterns in 2035-2044. The GHG-driven pattern is very strong in 2045-2054, with all SSPs having significantly larger temperature increases than SSP1-1.9 over Asia and East Asia, and SSP3-7.0 and SSP5-8.5 having significantly larger anomalies than SSP1-1.9 over South Asia.

2045-2054, Asia: SSP1-1.9 75th percentile = 1.59650, SSP2-4.5 median = 1.80856, SSP3-7.0 median = 1.67072, SSP5-8.5 median = 2.26456

2045-2054, East Asia: SSP1-1.9 75th percentile = 1.62415, SSP2-4.5 median = 1.90530, SSP3-7.0 median = 2.00293, SSP5-8.5 median = 2.27838

2045-2054, South Asia: SSP1-1.9 75th percentile = 1.21997, SSP3-7.0 median = 1.26886, SSP5-8.5 median = 1.45309

**The statements "The influence of aerosol is more clearly seen in regional mean precipitation than regional mean temperature,..."**

The statement that the influence of aerosol is more clearly seen in regional mean precipitation than regional mean temperature relates to the longer persistence of the aerosol-driven pattern across the SSPs in regional-mean precipitation than regional-mean temperature. As discussed above, we do not require significant differences between SSPs to identify an aerosol-driven signal as the physical interpretation of the result is more informative.

**...and, "Over Asia, the largest mean precipitation increase relative to 1980-2014 occurs in SSP1-1.9 for 2025-2034 and 2035-2044," both do not pass the significance test of Figure 2.**

The full phrase from the manuscript is: "Over Asia, the largest mean precipitation increase relative to 1980-2014 occurs in SSP1-1.9 for 2025-2034 and 2035-2044. The smallest precipitation increases are seen in SSP3-7.0 during these periods. Increases in SSP2-4.5 lie between those in SSP3-7.0 and SSP1-1.9." This is a description of the aerosol-driven pattern shown in Figure 2 (now Figure 3). In further support of this statement, the anomalies in SSP3-7.0 are significantly smaller than those in SSP1-1.9 for all periods shown in the figure, and those from SSP2-4.5 are significantly smaller than SSP1-1.9 for 2025-2034. We have added a note about the significant differences to the manuscript.

2025-2034, Asia: SSP1-1.9 25th percentile = 0.228189, SSP2-4.5 median = 0.224996, SSP3-7.0 median = 0.187030

2035-2044, Asia: SSP1-1.9 25th percentile = 0.234502, SSP3-7.0 median = 0.180172

2045-2054, Asia: SSP1-1.9 25th percentile = 0.238710, SSP3-7.0 median = 0.217290

**Is this because they are referring to the mean instead of the median?**

Here, 'mean precipitation' is used in reference to the regional-mean precipitation for Asia. Figure 8 shows the CMIP6 median and interquartile range of this quantity. We have specified 'regional-mean' and 'model-mean' in the revised manuscript for clarity.

**Figure 11 does support the statements made in discussion of Fig 7, but there is no indication of significance or robustness in that figure.**

The inclusion of both precipitation contours and wind vectors mean that this figure would be difficult to read if we also included an indication of significance or robustness. A figure showing just the precipitation anomalies relative to SSP1-1.9, and an indicator of their robustness (70% of models agree on the sign of the anomaly), is now included in the supporting information (Supplementary Figure 13). We have modified the text to note that the anomalies are robust. For consistency, we have also included the equivalent figure for the temperature anomalies shown in Figure 8 (now Figure 9) as new Supplementary Figure 12.

**With the presence of outliers, it is hard to know whether the mean values are biased in Figure 11.**

As noted in our previous response (and the previous version of the manuscript at line 13, page 8), the outliers in Figures 6 and 7 (now 7 and 8) arise because some models have relatively large or small regional-mean precipitation anomalies relative to the present-day. However, the relative differences between the SSPs for these models are not necessarily unusual compared to the other models. This is most easily seen in the South Asian case (Figure 7b, now 8b). For each time period there are 6 points that lie far below the interquartile range. The lowest two points (seen for SSP2-4.5 and SSP5-8.5) are from one model, while the four points above those are for another model. Although these points are all far below the CMIP6 median value, for each of the two outlier models the differences between the SSPs are comparable in size to the equivalent differences for the CMIP6 median. Thus, we do not expect such outliers to have an influence in Figure 11 (now Figure 12), where we show anomalies relative to SSP1-1.9.

The most striking outliers in Figure 7 (now Figure 8) are the high outliers for Asia-mean precipitation and low outliers for South-Asia-mean precipitation anomalies. The models in question are:

- High outlier for Asia (in 10/12 cases): CAMS-CSM1-0

- First low outlier for South Asia (in 6/12 cases): UKESM1-0-LL

- Second low outlier for South Asia (in 6/12 cases): EC-Earth3-Veg

While not a prominent outlier, IPSL-CM6A-LR consistently simulates the smallest anomaly in Asia-mean precipitation relative to the present-day.

We have redrawn Figure 11 (now Figure 12) with each of these models removed (Response Figures 1 to 3). However, note that EC-Earth3-Veg was not included in the original Figure 11 (now Figure 12) as it did not have precipitation data available for all SSPs, so an equivalent figure is not shown here for this model. None of these models that appear as outliers in Figure 7b affect the pattern of relative differences between SSPs shown in Figure 11 (now Figure 12).

**I also want to bring up the issue of land use land cover change (LULCC) again. The authors cite a paper (Singh et al. 2019b) that, along with additional references made within that paper, highlights the potential for LULCC to be just as important as other forcing agents on regional scales. The authors note this, but only show that globally LULCC is much smaller than aerosol and GHG forcing. The authors do bring up the regional impact LULCC may have in discussion of the South Asian results, but there is no discussion of LULCC with respect to the Asian or East Asian regions. Without calculating regional forcings, how are we supposed to know what the pattern of LULCC should look like among the different pathways. What if the LULCC forcings bring about the same pattern of responses in precipitation as do aerosols? While the model output may not allow for these calculations to be made reliably, it still merits additional discussion within the manuscript.**

We now include maps of historical Asian ERF for LULCC, anthropogenic aerosols, and total anthropogenic forcing for the CMIP6 multi-model mean as a new Figure, Figure 2, and show the corresponding maps for the individual models in the Supporting Information (Supplementary Figures 1-4). The same colour scale is used for each driver (and each model in the Supporting Information) to facilitate comparison. The CMIP6 mean is calculated using only the models where data is available to perform the necessary calculation for each driver. These models are indicated in Table 2 of the manuscript. Over land within the Asian domain, aerosol forcing is generally more than 5x the LULCC forcing. GFDL-CM4 has the largest LULCC forcing of the models we consider. Even in this model, the LULCC forcing over India is about 0.25x the aerosol forcing, and an even smaller factor over China. In addition to being larger in magnitude, the aerosol forcing is much more spatially extensive over land and ocean, and thus more likely to induce the type of continental-scale changes discussed in this manuscript.

[Figure]

**Figure 1.** Main text Figure 11 (now Figure 12) with CAMS-CSM1-0 removed. Hatching shows where at least 70% of models agree on the sign of the anomaly.

Considering total anthropogenic ERF shows large areas of Asian land experiencing negative radiative forcing, in patterns consistent with the anthropogenic aerosol forcing, although the global mean forcing is positive. Thus, based on the historical forcing, it seems unlikely that LULCC will produce Asian summer monsoon responses of a comparable magnitude to anthropogenic aerosols and greenhouse gases.

Both local and global aerosol forcing has a strong control on the Asian summer monsoon. The hemispheric asymmetry in aerosol forcing means that it is more efficient at driving shifts in the ITCZ and the summer monsoon circulation compared to homogeneous forcings. While there is considerable uncertainty in the magnitude and distribution of aerosol forcing across the CMIP6 models, this hemispheric asymmetry is a robust feature that is not present in the LULCC forcing (as can be seen in Response Figures 4 and 5). This difference further reduces the likelihood that LULCC has a comparable influence on the Asian summer monsoon to anthropogenic aerosol changes.

It is not possible to calculate the ERF for the SSPs, as the necessary timeslice experiments were not performed as part of CMIP6. This precludes a direct comparison of the forcing from LULCC and anthropogenic aerosols over the time period discussed in the manuscript. However, given the historical ERFs, we can speculate about the relative magnitude of the future forcing from LULCC and anthropogenic aerosols based on 1850-2100 emission and LULCC timeseries, which we now include in the Supporting Information (Supplementary Figure 5). Generally, the difference between the extreme SSPs in 2050 is comparable to the difference between 2014 and 1850 for aerosol and precursor emission changes, and LULCC. This makes it a reasonable first order assumption that the future influence of these climate drivers on the Asian summer monsoon will be comparable to the magnitude of their historical influence. Taken together, the historical and future changes in LULCC and anthropogenic aerosol both suggest that the Asian summer monsoon response to aerosol changes between the present day and 2050 are likely to be much larger than those due to LULCC, even on a regional scale. This statement is not necessarily a disagreement with the Singh et al. paper. It is likely a reflection of different usage of the word 'regional'. In our manuscript,

[Figure]

**Figure 2.** Main text Figure 11 (now Figure 12) with IPSL-CM6A-LR removed. Hatching shows where at least 70% of models agree on the sign of the anomaly.

we use 'regional' to refer to country-continental-scales, and 'regional' in the context of LULCC, including in the Singh paper, seems to refer to smaller spatial scales.

We now use these additional figures to support a slightly expanded discussion of the potential role of LULCC in the manuscript. However, the evidence available to us suggests that LULCC is a second-order consideration for near-term changes
5  in monsoon precipitation. Expanding the scope of the manuscript to include too much detail about LULCC would detract from the aerosol focus of the paper.

**"Comparing the SSP2-4.5aer and SSP2-4.5 responses (Figure 12 vs. Figure 13 for MIROC6...) shows that aerosol largely acts to offset the GHG-driven response, rather than determining the overall pattern of the response." How do we understand this by comparing these two figures? Many of the anomalies are the same sign as the eyeball-interpolated**
10  **trend in the SSP2-4.5 figures... so why does that make the SSP2-4.5aer response an offsetting effect for GHGs?**

Figure 12 (now Figure 13) shows the precipitation change from aerosol changes, and Figure 13 (now Figure 14) shows the total precipitation response. Figure 12 (now Figure 13) shows aerosol driving a precipitation decrease over northeast India, Bangladesh, and Burma/Myanmar. This anomaly is also present in Figure 13 (now Figure 14), where the expected response to a GHG increase alone would be a precipitation increase, suggesting that aerosol is offsetting the GHG-driven response. This
15  feature of drying in the vicinity of the northern coast of the Bay of Bengal is present in SSP2-4.5 in most models, as shown in Supplementary Figure 16. It can be seen in Figure 11 (now Figure 12) that the anomalous drying here relative to SSP1-1.9 is also larger in SSP2-4.5 and SSP5-8.5 (and robust across models) compared to SSP3-7.0 (where it is not robust across models), consistent with the differences in the regional-mean anomalies (Figure 7, now Figure 8).

There is a need for future work to explore the mechanisms underlying the spatial pattern of this response. We don't seek to
20  explain it here, but rather highlight it as a potentially interesting feature of the response in cases where local emissions continue to increase in future. We now stress this more clearly in the manuscript.

[Figure]

P responses in relative to SSP1-1.9

**Figure 3.** Main text Figure 11 (now Figure 12) with UKESM1-0-LL removed. Hatching shows where at least 70% of models agree on the sign of the anomaly.

**This dipole concern in Figures 12 and 13 leads into a bigger question about the dipole. Knowing that the responses over South Asia (Figure 7) are not in agreement with the aerosol driven paradigm of Figure 2, how should the dipole pattern be interpreted?**

This deviation of the dipole response from the aerosol-driven paradigm of Figure 2 (now Figure 3) is precisely what makes
5  this feature interesting, and worthy of further discussion in the manuscript, beyond the regional mean values shown in Figure 7 (now Figure 8). We have expanded our discussion of the dipole pattern in the revised manuscript.

The paradigm presented in Figure 2 (now Figure 3) is based on seeing aerosol decreases in all scenarios globally, and for most regions. However, this is not the case in South Asia, where continued aerosol increases are projected until 2030-2050, depending on the scenario. If the South Asian summer monsoon response to aerosol was determined only by South Asian
10  aerosol changes, then we could have presented a different paradigm for this region consistent with the local emission changes, and tried to identify it in the CMIP6 data. However, there is a large body of literature that suggests that the South Asian summer monsoon response to aerosol changes is sensitive to changes in *both* local and remote aerosol, with varying conclusions about the relative importance of the different emission regions (e.g. Guo et al., 2016; Undorf et al., 2018). In the case where remote emission changes are the dominant driver, we would expect to see the aerosol-driven precipitation pattern presented in Figure
15  2 (now Figure 3) in the South-Asian-mean precipitation response. The fact that we do not see this suggests that there is an important role for local emission changes, and the potential for regional policy decisions to affect regional climate. This is the first study to demonstrate that there is a fingerprint of the observed Asian aerosol dipole (Samset et al., 2019) in precipitation projections, and suggests that further study into the dynamical mechanisms underlying this response is warranted. In addition, the emergence of the aerosol emissions dipole in observations will need continued monitoring in order to inform assessments
20  on the realism of the various SSP scenarios for the near-term and mid-term future.

[Figure]

**Figure 4.** 1850-2014 ERF (annual mean) due to anthropogenic aerosols for each model used in this study (where data is available). The number in the top right of each panel shows the global mean forcing in W m$^{-2}$.

[Figure]

**Figure 5.** 1850-2014 ERF (annual mean) due to land use and land cover changes for each model used in this study (where data is available). 
[revised manuscript text omitted]